# SpikeCLIP: A Contrastive Language-Image Pre-trained Spiking Neural Network

## Abstract

Spiking neural networks (SNNs) have demonstrated the capability to achieve comparable performance to deep neural networks (DNNs) in both visual and linguistic domains while offering the advantages of improved energy efficiency and adherence to biological plausibility. However, the extension of such single-modality SNNs into the realm of multimodal scenarios remains an unexplored territory. Drawing inspiration from the concept of contrastive language-image pre-training (CLIP), we introduce a novel framework, named SpikeCLIP, to address the gap between two modalities within the context of spike-based computing through a two-step recipe involving "Alignment Pre-training + Dual-Loss Fine-tuning". Extensive experiments demonstrate that SNNs achieve comparable results to their DNN counterparts while significantly reducing energy consumption across a variety of datasets commonly used for multimodal model evaluation. Furthermore, SpikeCLIP maintains robust performance in image classification tasks that involve class labels not predefined within specific categories.

## 1 Introduction

While modern deep neural networks achieve impressive performance on a variety of image, audio, and language tasks and sometimes even perform better than humans, their substantial energy requirements have become a subject of increasing scrutiny. Representative examples like ChatGPT (OpenAI, 2022) and GPT-4 (OpenAI, 2023) have exhibited significant energy consumption, especially when engaged in complex reasoning tasks. Consequently, the energy-efficient advantage of SNNs is garnering escalating interest and recognition within the machine-learning community. Emerging as the third generation of neural networks (Maass, 1997), SNNs have drawn increasing attention due to their biological plausibility, event-driven nature, rapid inference capabilities, and efficient energy utilization (Pfeiffer & Pfeil, 2018; Roy et al., 2019). Utilizing SNNs in the development of extensive computational models offers the potential for significant energy efficiency and subsequent cost reductions in the implementation of large-scale applications, thereby promoting further advancements with such a computational paradigm.

Within the realm of computer vision, SNNs have achieved great success in image classification (Cao et al., 2015; Diehl et al., 2015; Rueckauer et al., 2017; Hu et al., 2018; Yin et al., 2020; Fang et al., 2021; Zhou et al., 2023a;b). Among them, a series of works by Spikingformer (Zhou et al., 2023a;b), inspired by the Vision Transformer (ViT) (Dosovitskiy et al., 2010), have proposed effective SNNs architectures grounded in hardware feasibility. In contrast to their application in computer vision, the utilization of SNNs in natural language processing remains relatively limited (Rao et al., 2022; Lv et al., 2022; Zhu et al., 2023b), with only a handful of studies exploring the potential of SNNs in text processing tasks. For example, Lv et al. (2022) proposed a TextCNN-based SNN to attempt to complete the task of text classification, despite the large performance difference with the Transformer-based language model.

Previous works on SNNs largely targeted single-modality input representations using spikes. However, the exploration of extending SNNs to multimodal contexts remains uncharted territory. To address this gap, we introduce SpikeCLIP, inspired by the dual-stream CLIP trained via contrastive learning (Radford et al., 2021). Through SpikeCLIP, we evaluated the feasibility and potential of using the spike paradigm to handle multimodal tasks.

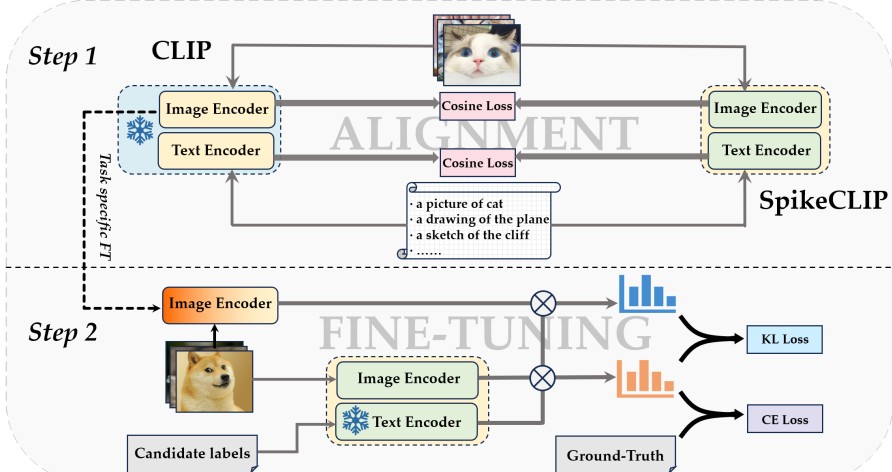

Figure 1: An illustration of a two-step recipe of "Alignment Pre-training + Dual-Loss Fine-tuning". During the first step, SpikeCLIP learns to generate high-quality representations of both images and text; In the second step, the image encoder of SpikeCLIP undergoes further fine-tuning on the downstream dataset by a joint loss function including the KL loss and the CE loss.

SpikeCLIP is the first multimodal SNN, trained using the method of "Alignment Pre-training + Dual-Loss Fine-tuning". Specifically, we initially maximize the cosine similarity between the output representations of CLIP and SpikeCLIP, both image-side and text-side, utilizing a large pre-training dataset. This allows SpikeCLIP to generate universal representations of images and text, a process called "Alignment Pre-training". Subsequently, to enhance SpikeCLIP's performance on targeted downstream datasets, we undertook the "Dual-Loss Fine-tuning" process, emphasizing the optimization of Kullback-Leibler divergence (KL) loss and Cross-Entropy (CE) loss. The KL loss is calculated based on the class probability distribution that SpikeCLIP and the task-specific fine-tuned CLIP yield during the classification, while the CE loss is determined by contrasting the class probability distribution produced by SpikeCLIP against the actual labels (see Figure 1 for details). Similar to CLIP, SpikeCLIP possesses zero-shot learning ability (Table 2) and has the flexibility to circumvent the constraints associated with fixed labels in classification tasks (Table 4).

The contribution of this study can be summarized as follows:

- We have demonstrated for the first time that SNNs can perform feature extraction and alignment across multiple modalities through spiking trains. Based on the findings, we propose a cross-modal SNN, named SpikeCLIP, which performs well in cross-modal alignment between images and text.
- A training method is also proposed with a novel "Alignment Pre-training + Dual-loss Fine-tuning" strategy. With pre-trained SpikeCLIP, we make it possible to efficiently fine-tune SpikeCLIP on subsequent datasets without necessitating initialization from scratch for a new dataset.
- SpikeCLIP not only exhibits competitive performance when compared to existing single-modal SNNs but also empowers the spiking computing paradigm to overcome the constraints of the fixed label quantification intrinsic to image classification.

## 2 RELATED WORK

Unlike traditional Artificial Neural Networks (ANNs), SNNs employ spikes in a stimulus time window (time step, denoted $T$) for information processing, demonstrating biological plausibility, event-driven nature, rapid inference capabilities, and efficient energy utilization (Pfeiffer & Pfeil, 2018; Roy et al., 2019). In recent years, there has been substantial attention on SNNs, resulting in numerous studies dedicated to discovering more efficient architectures and training methods.

In computer vision (CV), a lot of progress has been made in SNNs. Cao et al. (2015) demonstrated the feasibility of applying the weights of Convolutional Neural Networks (CNNs) to SNNs, which

have similar architectures as the original CNNs. This approach exemplifies the transformation of ANNs into SNNs using weight conversion. Similarly, Wang et al. (2022) devised strategies incorporating signed neurons and memory functionalities to counteract the performance decline observed during the ANN-to-SNN conversion. Furthermore, Bu et al. (2023) implemented a quantized clip background shift activation function in initial ANNs, surpassing traditional ReLU functions and mitigating performance degradation in the ANN-to-SNN transition. In contrast to the method of constructing SNNs from ANNs, some studies employ surrogate gradients to directly train SNNs during backpropagation. For instance, Wu et al. (2018) proposed a Spatio-Temporal Backpropagation (STBP) training framework, introducing an approximate derivative to address the non-differentiable issue related to spiking activities. Expanding on STBP, Zheng et al. (2021) proposed a Threshold Correlated Batch Normalization (tdBN) method, enabling the creation of deeper layers within SNNs by utilizing emerging spatiotemporal backpropagation techniques. Additionally, the innovative approach by Zhou et al. (2022) introduced Transformer-based architectures to SNNs, marking significant advancements in image classification performance. Subsequent enhancements to this groundbreaking model are documented in Zhou et al. (2023a;b), contributing to the continuous refinement and improvement of performance in this field.

In Natural Language Processing (NLP), the exploration of SNNs is relatively nascent. A few seminal works have marked progress in this domain. For instance, Lv et al. (2022) pioneered text classification by transmuting word embeddings into spike trains. Additionally, Bal & Sengupta (2023) innovated an SNN architecture analogous to BERT through knowledge distillation, as elucidated by Hinton et al. (2015). Moreover, Zhu et al. (2023b) delved into the SNNs for text generation, utilizing an architecture analogous to Recurrent Neural Networks (RNNs). In multimodal processing, a myriad of prominent multimodal models grounded in ANNs have been developed, with examples like OSCAR (Li et al., 2020) and SimVLM (Wang et al., 2021) representing single-stream architectures, and CLIP (Radford et al., 2021) and WenLan (Huo et al., 2021) exemplifying dual-stream architectures. However, multimodal SNNs remain largely unexplored due to their challenging training and generally inferior performance compared to ANN counterparts. Nevertheless, drawing inspiration from the pioneering efforts documented in Zhou et al. (2022; 2023a;b), there emerges a promising avenue for the conception of multimodal models rooted in SNNs, taking cues from CLIP (Radford et al., 2021). CLIP utilizes a combined image and text encoder, trained through contrastive learning from extensive image-text pairs. Inspired by CLIP, our SpikeCLIP demonstrates for the first time that SNNs also perform well in feature alignment between images and text.

## 3 METHOD

Inspired by CLIP (Radford et al., 2021), we perform image classification by evaluating the semantic similarity between visual and textual representations. This methodology incorporates semantically supervised information through the alignment of image and text modalities, thereby obviating the need for explicit classification within the model. Given the strong image representation ability of SNNs (Zhou et al., 2023a;b) and the demonstrated success of spiking representations for text embeddings (Lv et al., 2022), we posit that text information encoded in spiking signals can synergistically complement spiking image representations to accomplish multimodal tasks. In the SpikeCLIP architecture, the image encoder is based on Spikingformer (Zhou et al., 2023b), while the text encoder is a Spiking Multi-Layer Perceptron (S-MLP).

During the pre-training, our primary focus is to optimize the cosine similarity between the output representations produced by both the image and text encoder of CLIP and SpikeCLIP, as described in Equation 3. This process facilitates the alignment of general representations between SpikeCLIP and CLIP. Before fine-tuning SpikeCLIP, a CLIP is fine-tuned on a specific dataset. The fine-tuned CLIP serves to guide the modification of SpikeCLIP's probability distribution before classification, as articulated by the loss function specified in Equation 4. Additionally, SpikeCLIP receives supervision from ground-truth labels, as captured in the loss function presented in Equation 5. During inference, SpikeCLIP is fed an image and several candidate text labels associated with it. After calculating the cosine similarity between the image representation and various text representations, the text label with the highest cosine similarity is selected as the best output. The overall architecture of SpikeCLIP is illustrated in Figure 2. In the following, we start with an overview of spiking neurons, then explore the architecture of SpikeCLIP, and finally discuss the training methodology used.

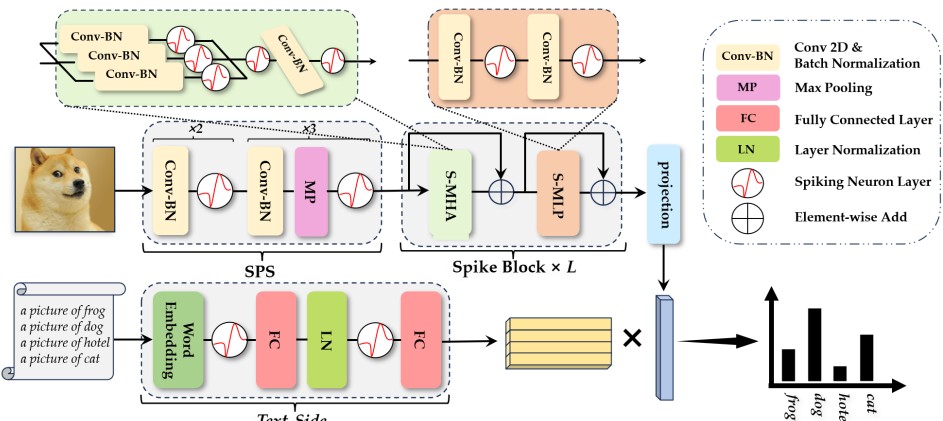

Figure 2: The architecture of SpikeCLIP. The image processing component comprises a Spiking Patch Splitting (SPS) layer, multiple Spike Blocks, and a projection layer. Within each Spike Block, there is a Spiking Multi-Head Attention (S-MHA) module as well as a Spiking Multi-Layer Perceptron (S-MLP) module. SpikeCLIP's text processing component integrates a Word Embedding layer along with an MLP-based module. Communication between these individual modules is facilitated through binary code, leading to lower energy consumption.

### 3.1 INTEGRATE-AND-FIRE NEURON

Leaky Integrate-and-Fire (LIF) neurons are extensively utilized within SNNs to construct the Spiking Neuron Layer (shown in Figure 2), and serve a role analogous to activation units in ANNs. Different from the activation units in ANNs, LIF neurons function akin to a Heaviside step function as the networks propagate forward, wherein all floating-point numbers within the data stream are transformed into binary integers, either $0$ or $1$. LIF neurons operate on the weighted sum of inputs. The membrane potential of the neuron $U_t$ is affected by these inputs at a given time step $t$. The neuron will produce a spike $S_t$, once its membrane potential exceeds the threshold $U_{\mathrm{thr}}$, as follows:

$$S_t = \begin{cases} 1, & \text{if } U_t \geq U_{\mathrm{thr}}; \\ 0, & \text{if } U_t < U_{\mathrm{thr}}. \end{cases} \tag{1}$$

The dynamic equation governing the membrane potential of LIF neurons is presented as follows:

$$U_t = I_t + \beta U_{t-1} - S_{t-1} U_{\mathrm{thr}}, \quad I_t = W X_t \tag{2}$$

where $U_t$ and $U_{t-1}$ are the membrane potentials at the time of $t$ and $t-1$ respectively. $I_t$ signifies the weighted sum of inputs at time $t$, while $\beta$ represents the rate of membrane potential decay. $W$ comprises a set of learnable weights. Furthermore, the expression $S_{t-1} U_{\mathrm{thr}}$ encapsulates the logic governing the reset of the membrane potential.

### 3.2 ARCHITECTURE

The architecture of SpikeCLIP is shown in Figure 2. The model is composed of two primary components: an image encoder and a text encoder. Because Spikingformer (Zhou et al., 2023a) is not only based on a Transformer architecture (like CLIP) but also achieves optimal performance in image classification tasks, we chose to use it as the base model for the image encoder of SpikeCLIP. In addition, the image encoder combines outputs across multiple time steps through the use of Time-Steps Weight, which is an algorithm design that takes into account the interaction of spike signals with different time steps.(see Appendix A.1 for the rationale behind this design choice). As for the text encoder of SpikeCLIP, after evaluating the performance of Transformer-based and Multi-Layer Perceptron (MLP)-based architectures, we chose a simpler MLP-based architecture as the text encoder for SpikeCLIP (a comparative analysis can be found in Appendix A.3).

### 3.3 PRE-TRAINING AND FINE-TUNING

We introduce a two-step training method of "Alignment Pre-training + Dual-Loss Fine-tuning" to align the semantic spaces of image and text modalities. For convenience, we will refer to a conventional

CLIP as $C$. First, we use $C$ to help align the output representations of the image and text sides of SpikeCLIP in general. This step enables SpikeCLIP to generate high-quality representations for images and text, as well as possess some zero-shot learning ability. Then, we fine-tune $C$ on a downstream dataset. We represent the image encoder of the fine-tuned $C$ as $C_{fv}$, and $C_{fv}$ is used as a teacher model when fine-tuning the SpikeCLIP image encoder. In "Dual-Loss Fine-tuning", SpikeCLIP receives supervision from the teacher model and the ground-truth labels through KL Loss and CE Loss, respectively.

### 3.3.1 LANGUAGE-IMAGE PRETRAINING

In the following, the image encoder and text encoder of $C$ will be referred to as $C_l$ and $C_v$. We will also designate the image and text encoder of SpikeCLIP as $SC_v$ and $SC_l$. The datasets used for pre-training SpikeCLIP image and text encoders are denoted as $D_{\text{img}}$ and $D_{\text{txt}}$.

Diverging from the direct application of contrastive training, which may result in gradient vanishing or exploding, we adopt the idea of KD to align the image encoder of SpikeCLIP using spike signals with the image representation generated by the CLIP image encoder. The same alignment approach is applied to the text encoder. This design tackles the challenge of directly aligning two types of pulse signals by introducing the floating-point representations generated by CLIP as a "bridge."

The specific operations are as follows: during the pre-training of $SC_v$ (or $SC_l$), for any given image (or text) $x_i$ in a dataset $D_{\text{img}}$ (or $D_{\text{txt}}$) of size $N$, two latent space vectors $v_i$ and $\hat{v}_i$ are generated after the image passes $C_v$ and $SC_v$ (or the text passes $C_l$ and $SC_l$), respectively. The objective of the pre-training is to maximize the cosine similarity between $v_i$ and $\hat{v}_i$. The loss function is formulated as follows, where $N$ is the number of training instances:

$$L = \frac{1}{N} \sum_{i=1}^{N} (1 - \frac{v_i \cdot \hat{v}_i}{\|v_i\| \cdot \|\hat{v}_i\|}) \tag{3}$$

### 3.3.2 FINE-TUNING GUIDED BY DUAL LOSS

We perform fine-tuning by optimizing both the KL Loss and the CE Loss on a downstream dataset (denoted $D_{\text{down}}$). As in the work by Kingma & Welling (2013) and Zhu et al. (2023a), we use the two losses to construct a joint loss, which enables SpikeCLIP to automatically consider both the KL loss function and the CE loss function when optimizing the joint loss function. Among them, the CE loss guarantees the consistency of SpikeCLIP with the real labels. On this basis, because of the loss caused by the inherent spike signals of SpikeCLIP, we ensure the consistency of SpikeCLIP with the task-specific fine-tuned CLIP by applying the KL loss as a penalty.

The model will try to find a balance and ultimately minimize the sum of these two loss functions. We describe the fine-tuning process in detail below.

Before fine-tuning SpikeCLIP, we need a conventional CLIP fine-tuned on the dataset $D_{\text{down}}$, and its image encoder is $C_{fv}$, which is used as a teacher model. Additionally, since the architecture of SpikeCLIP's text encoder ($SC_l$) is relatively simpler than that of the image encoder ($SC_v$), and the dataset ($D_{\text{txt}}$) used to train the text encoder is sufficient, the text encoder has been trained enough. Therefore, we freeze the parameters of the text encoder during fine-tuning to prevent its parameters from being updated (refer to Appendix A.3 for details). Then, we construct a label text set (denoted $Candidate\ labels$ in Figure 2) containing $M \times k$ text instances by combining the $M$ labels and the corresponding $k$ templates from dataset $D_{\text{down}}$. After feeding $Candidate\ labels$ to $SC_l$, we obtain $M$ text representations with dimension $d$ for classification, called $Candidates$, similar to the "potential text pairings" in CLIP (Radford et al., 2021).

During the fine-tuning, any image $x_i$ from $D_{\text{down}}$ is fed separately into $SC_v$ and $C_{fv}$, outputting two distinct latent $v_i$ and $\hat{v}_i$ of dimension $d$ respectively. Subsequently, matrix multiplication is performed with $v_i$ and $\hat{v}_i$ respectively against $Candidates$, obtaining two class probability distributions $pre_i$ and $\hat{pre}_i$. We guide $pre_i$ with $\hat{pre}_i$ through minimizing the KL Loss, ensuring that the classification probability distribution of SpikeCLIP does not deviate too much from its corresponding CLIP during the fine-tuning. This constraint is based on knowledge distillation (Hinton et al., 2015), with CLIP as a teacher, guiding $SC_v$ to update parameters in a more stable direction. The CE Loss is derived from

$pre_i$ and ground-truth label $y_i$. In conjunction with KL Loss, CE Loss enhances the efficiency of SpikeCLIP's fine-tuning on the downstream dataset (refer to Table 3 for details).

The KL loss, CE loss, and Joint loss are defined below:

$$\text{KLDivLoss} = \frac{1}{N} \sum_{i=1}^{N} \sum_{j=1}^{M} \hat{pre}_{ij} \log \left( \frac{\hat{pre}_{ij} + \epsilon}{pre_{ij} + \epsilon} \right) \tag{4}$$

$$\text{CELoss} = -\frac{1}{N} \sum_{i=1}^{N} \sum_{j=1}^{M} y_{ij} \log(pre_{ij}) \tag{5}$$

$$\text{JointLoss} = \text{KLDivLoss} + \alpha \cdot \text{CELoss} \tag{6}$$

where $N$ is the number of training instances for the downstream dataset, $\epsilon$ is a small constant, such as $\epsilon = 1 \times 10^{-10}$, set for numerical stability and to avoid division by zero, and $\alpha$ is a hyperparameter and defaults to 1.

## 4 EXPERIMENTS

We conducted four experiments to thoroughly evaluate SpikeCLIP. Section 4.2 presents its CIFAR dataset performance and zero-shot learning ability. In Section 4.3, we extensively studied pre-training's importance, the impact of pre-training data, and the influence of loss functions during fine-tuning. Section 4.4 evaluates SpikeCLIP's modality alignment, while Section 4.5 analyzes its energy efficiency. Dataset details are in Section 4.1, and experimental settings are in Appendices A.2 and A.3.

### 4.1 DATASET

We used the ImageNet-1k dataset (Russakovsky et al., 2015) for pre-training and the following six datasets as downstream datasets: CIFAR10 (Krizhevsky, 2009), CIFAR100 (Krizhevsky, 2009), Flowers102 (Nilsback & Zisserman, 2008), OxfordIIITPet (Parkhi et al., 2012), Caltech101 (Fei-Fei et al., 2004), and STL10 (Coates et al., 2011). These datasets are well-known and have varying numbers of labels for image classification tasks. Additionally, we constructed a new dataset ($D_{\text{txt}}$), from labels and templates of all datasets used to assess CLIP, containing 115,708 text entries. The dataset, used for pre-training SpikeCLIP's text encoder, encapsulates a wide array of standard text labels pertinent to image classification tasks (See Appendix A.4 for details).

### 4.2 IMAGE CLASSIFICATION

In this section, we conduct two experiments: First, we compare the performance difference between SpikeCLIP and the previous models trained on either single-modal or multi-modal data. Secondly, since we are unable to access the complete dataset used to pre-train CLIP as it is not publicly available, we utilize an ANN counterpart to SpikeCLIP, named ScratchCLIP, for comparative experiments with SpikeCLIP. To ensure fairness, ScratchCLIP's image encoder adopts the Transformer architecture, and its text encoder uses the MLP architecture. While its parameters are similar to SpikeCLIP's, it lacks spiking neurons and processes data in floating-point form. Moreover, both models were pre-trained and fine-tuned under the same conditions.

#### 4.2.1 RESULTS ON CIFAR

The accuracy on CIFAR achieved by SpikeCLIP is reported in Table 4.2.1, compared to baseline models. In Table 4.2.1, Hybrid training (Rathi et al., 2020), Diet-SNN (Rathi & Roy, 2020), STBP (Wu et al., 2018), STBP NeuNorm (Wu et al., 2019), TSSL-BP (Zhang & Li, 2020), STBP-tdBN (Zheng et al., 2021), TET (Deng et al., 2022), TEBN (Duan et al., 2022) and Spikingformer (Zhou et al., 2023b) are single-modality SNNs. For ANNs, ViT (ViT-B/16 [1]) (Dosovitskiy et al., 2010) is one of the top-performing single-modality ANNs, while CLIP (Dosovitskiy et al., 2010) is one

---

[1] https://github.com/google-research/vision_transformer

Table 1: Accuracy results on CIFAR datasets. SpikeCLIP achieves accuracy of 94.48% and 77.69% on CIFAR10 and CIFAAR100 respectively, surpassing all single-modality SNNs except Spiking-former (with a small performance drop of 1.47% and 2.68%). The best and second-best results of SNNs and ANNs are highlighted with bold fonts, as well as their performance gaps indicated by "Gap (Accuary)". Note that the performance gap between SpikeCLIP and its single-modality state-of-the-art model (i.e., Spikingfomer) is much less than that between the conventional CLIP and ViT (SOAT traditional ANN on CIFAR datasets).

| | Method | Param (M) | Time Step | CIFAR 10 | CIFAR 100 | Gap (Accuracy) |
|---|---|---|---|---|---|---|
| **SNNs** | Hybrid training | 9.27 | 125 | 92.22 | 67.87 | |
| | Diet-SNN | 0.27 | 10/5 | 92.54 | 64.07 | |
| | STBP | 17.54 | 12 | 89.83 | −− | |
| | STBP NeuNorm | 17.54 | 12 | 90.53 | −− | −− |
| | TSSL-BP | 17.54 | 5 | 91.41 | −− | |
| | STBP-tdBN | 12.63 | 4 | 92.92 | 70.86 | |
| | TET | 12.63 | 4 | 94.44 | 74.47 | |
| | TEBN | − | 4 | 95.58 | 78.71 | |
| | Spikingformer | 9.32 | 4 | **95.95** | **80.37** | 1.47/**2.68** |
| | SpikeCLIP (ours) | 56.87 | 4 | **94.48** | **77.69** | |
| **ANNs** | ViT | 86.39 | 1 | **99.13** | **94.20** | **0.68**/4.50 |
| | CLIP | 149.6 | 1 | **98.45** | **89.70** | |

of the best-performing multimodal ANNs. According to the data in Table 4.2.1, it is evident that SpikeCLIP has a higher classification accuracy (94.48%/77.69%) than any other single-modality SNN on the CIFAR dataset, except for Spikingformer, which currently holds the top spot. However, it is worth noting that single-modality models tend to perform better than multi-modality ones, even in ANNs. As shown in the table, ViT, a single-modality model, outperforms CLIP on CIFAR10/100 by **0.68%/4.5%**. Therefore, it is reasonable to expect a performance gap (**1.47%/2.68%**) between SpikeCLIP and Spikingformer on CIFAR10/100 for SNNs.

Overall, the comparison between the two gaps described above illustrates the degree of performance of SpikeCLIP, which sets the benchmark for future multimodal SNNs on the same dataset.

### 4.2.2 ZERO-SHOT RESULTS

CLIP is trained using a large dataset composed of numerous image-text pairs, but this dataset is not open source and we cannot train SpikeCLIP with it. For evaluating the zero-shot learning ability of SpikeCLIP and its ANN counterpart, ScratchCLIP, we resort to using ImageNet-1k as the pre-training dataset for both, as ImageNet-1k is one of the largest image-text classification datasets available to us. To compare their zero-shot learning ability, SpikeCLIP and ScratchCLIP are evaluated on downstream datasets for accuracy after being trained for the same number of epochs on the ImageNet-1k dataset.

Table 2: Zero-shot classification results. CLIP is a pre-trained model (openai/clip-vit-base-patch16). ScratchCLIP is an ANN with a transformer on the image side and an MLP on the text side.

| Model | CIFAR 10 | CIFAR 100 | Flowers 102 | Caltech 101 | OxfordIIITPet | STL 10 | Avg |
|---|---|---|---|---|---|---|---|
| ScratchCLIP | 59.70 | 27.94 | 8.33 | 48.72 | 48.60 | 75.69 | 44.83 |
| SpikeCLIP | 58.03 | 26.66 | 9.02 | 48.28 | 44.89 | 77.79 | 44.11 |

*Note:* For comparison with SpikeCLIP: (a) ScratchCLIP's image encoder has four layers like SpikeCLIP; (b) In the image encoder of ScratchCLIP, a patch splitting layer with the same parameters as the SPS layer in SpikeCLIP is used to maintain the same parameter level as SpikeCLIP; (c) ScratchCLIP undergoes the same rounds of pre-training as SpikeCLIP on ImageNet-1k, followed by zero-shot classification on the downstream dataset.

According to the data presented in Table 4.2.2, SpikeCLIP has an average accuracy of 44.11% on downstream datasets. This is slightly lower than its ANN counterpart, ScratchCLIP, which has an average accuracy of 44.83%. However, the difference between the two is only 0.72%, which is negligible. Despite the fact that SpikeCLIP uses integer operations to conserve energy, which distinguishes it from ScratchCLIP, it still performs competitively under equivalent pre-training

conditions. Therefore, we can reasonably assume that SpikeCLIP's performance could be further improved with additional training data.

## 4.3 ABLATION EXPERIMENTS

We conducted some ablation experiments to investigate the impact of SpikeCLIP performance by the following three factors:

- Pre-training with large-scale dataset.
- The size of and the data distribution of datasets used for pre-training.
- Dual loss applied in fine-tuning stage.

Table 3: Ablation study. The top-performing results in each column are highlighted. **E1** reveals that pre-training with LSD significantly improves the model's classification performance on downstream datasets; **E2** affirms that optimizing both losses during fine-tuning yields the most significant performance boost.

|  | Setting | CIFAR 10 | CIFAR 100 | Flowers 102 | Caltech 101 | OxfordIIITPet | STL 10 | Avg |
|---|---|---|---|---|---|---|---|---|
| E1 | w/o LSD | 93.23 | 74.59 | 66.98 | 23.67 | 34.94 | 69.25 | 60.44 |
|  | w/ LSD | **94.48** | **77.69** | **86.07** | **82.31** | **67.18** | **89.48** | **82.89** |
| E2 | CE | 94.22 | 77.52 | 82.86 | 66.01 | 88.92 | 65.29 | 78.69 |
|  | KL | 94.20 | 77.42 | 81.76 | 65.95 | 89.58 | 62.72 | 78.61 |
|  | CE + KL | **94.33** | **77.68** | **82.97** | **66.34** | **89.59** | **86.47** | **82.90** |

**Pre-training with large scale dataset.** Previous single-modality SNNs could only be trained from scratch on new datasets when performing image classification tasks. This meant that for each specific downstream dataset, a different model needed to be trained, which was highly inefficient. However, our SpikeCLIP can effectively achieve zero-shot classification results on various downstream datasets through "Alignment Pre-training" and only requires fine-tuning on the downstream dataset to significantly improve classification performance. This is the first pre-training and fine-tuning paradigm based on the SNNs framework. To compare with the pre-training setup using a large-scale dataset (**LSD**), we completed the "Alignment Pre-training + Dual-Loss Fine-tuning" steps on all downstream datasets separately. As shown in E1 of Table 3, when pre-training is performed using LSD, the increase in accuracy ranged from $1.25\%$ to $58.64\%$, with an average improvement of $22.45\%$.

**Dataset Size and data distribution during pre-training.** Our SpikeCLIP has demonstrated impressive results on downstream datasets despite being pre-trained only on a limited dataset of ImageNet-1k. However, we believe that expanding the pre-training dataset could further enhance its performance. In pursuit of this hypothesis, we present the following discussions and experimental designs:

Generally, a model's performance improves with the amount of data it is trained on, and this can be measured by the size of the data volume and the similarity between the training and evaluation datasets. Larger amounts of data and more similar distributions between the two datasets typically lead to better evaluation results. Taking these factors into consideration, we establish gradients of data size and form three different data distribution groups for each size: Slightly-similar, Intermediate, and Dissimilar. Please refer to Appendix A.6 for more details. Figure 4.3 illustrates that SpikeCLIP follows these conclusions, which leads us to believe that training SpikeCLIP on larger and more varied datasets could result in even better performance.

**Dual-loss for fine-tuning.** During the fine-tuning stage, we utilize joint loss to update the parameters of $SC_v$, which includes two losses: the KL loss and the CE loss. The CE loss relies on the model's ground-truth labels to guide training, while the KL loss ensures that the model captures the ranking information of classification probabilities generated by $C_{fv}$. This dual-loss approach helps maintain weight stability during gradient updates, as demonstrated in E2 of table 3. Our hypothesis is confirmed as SpikeCLIP performance improves when both CE and KL loss functions are applied.

## 4.4 CROSS-MODAL IMAGE CLASSIFICATION

In this section, we demonstrate the effect of SpikeCLIP in aligning modality information between images and text into the same semantic space using two methods — Expanded Label Set (**ELS**) and Unseen Label Set (**ULS**). The implementation details of the two methods are detailed in Appendix

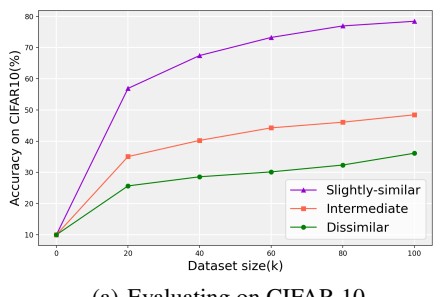
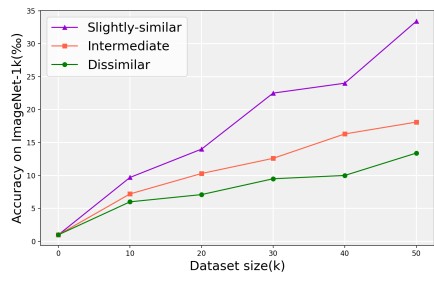

(a) Evaluating on CIFAR 10            (b) Evaluating on ImageNet-1k

Figure 3: The impact of dataset size and data distribution. The training data is sampled from various datasets, leading to differences in similarity between the training dataset and the evaluation dataset. (a) Slightly-similar: ImageNet-1k + CIFAR100 + CIFAR10; Intermediate: ImageNet-1k + CIFAR100; Dissimilar: ImageNet-1k. (b) Slightly-similar: CIFAR10 + CIFAR100 + ImageNet-1k; Intermediate: CIFAR10 + CIFAR100; Dissimilar: CIFAR10.

A.5. Compared to the baseline, both transformation methods have a low performance penalty. It's worth noting that this is the first time SNNs have achieved modal alignment in classification tasks without the constraint of fixed labels.

Table 4: Cross-modal image classification. In ELS, the dataset's label set is expanded to $N$ times the original, where $N \in \{2, 5, 8\}$. In ULS, unseen label words are used to replace the label set of the downstream dataset, according to a replacement ratio $\alpha$, where $\alpha \in \{20\%, 40\%, 80\%, 100\%\}$. Experimental results from both ELS and ULS strategies demonstrate that SpikeCLIP excels in achieving accurate image-text alignment and exhibits robustness in image classification tasks.

| Dataset | Baseline | ELS | | | ULS (Acc/Std) | | | |
|---|---|---|---|---|---|---|---|---|
| | | ×2 | ×5 | ×8 | 20% | 40% | 80% | 100% |
| CIFAR 10 | 94.33 | 94.33 | 94.33 | 94.32 | 94.33(0.028) | 94.32(0.033) | 94.22(0.017) | 94.18 |
| STL 10 | 89.59 | 89.59 | 89.59 | 89.45 | 89.45(0.008) | 89.20(0.127) | 87.42(0.504) | 87.64 |

## 4.5 ENERGY CONSUMPTION

We report in Table 5 the average firing rate of spiking neurons (**Firing Rate**), energy consumption (**Energy**), and energy reduction (**Energy Reduction**) rate of SpikeCLIP compared to ScratchCLIP on downstream datasets. The calculation methods are shown in Appendix A.7.

Table 5: Energy consumption. SpikeCLIP reduces energy consumption by 77.06% to 78.66% compared to its ANN counterpart.

| Dataset | CIFAR 10 | CIFAR 100 | Flowers 102 | Caltech 101 | OxfordIIIPet | STL 10 |
|---|---|---|---|---|---|---|
| Firing Rate(%) | 27.26 | 28.98 | 29.30 | 27.97 | 27.93 | 27.56 |
| Energy(mJ) | 3.17 | 3.37 | 3.41 | 3.25 | 3.25 | 3.21 |
| Energy Reduction | 78.66%↓ | 77.31%↓ | 77.06%↓ | 78.10%↓ | 78.13%↓ | 78.42%↓ |

## 5 CONCLUSION

This study has illustrated the capacity of Spiking Neural Networks (SNNs) to effectively capture multi-modal features and perform multi-modal classifications with remarkable proficiency, contingent upon the alignment of features from distinct modalities. We introduced SpikeCLIP, a novel multi-modal SNN architecture, underpinned by the innovative training approach termed "Alignment Pre-training + Dual-Loss Fine-tuning". SpikeCLIP exhibits impressive classification capabilities and also demonstrates promise under the setting of zero-shot learning. By successfully bridging the gap in the application of SNNs within multi-modal scenarios, this research serves as a fundamental stepping stone, laying the groundwork for prospective investigations in this field.

## REPRODUCIBILITY STATEMENT

The datasets used in the above experiments are all open source. In order to replicate the experiments in the section 4.2, 4.3, and 4.4, we have provided all the code and running scripts in the supplementary materials. We have also provided a README script that guides how to run the code. In addition, the project will be published on Github to provide experimental support.

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

## A APPENDIX

### A.1 THE ADDITION OF LEARNABLE TIME-STEPS WEIGHT(TSW) PARAMETERS

In previous SNNs, tensor values were averaged across different time steps ($T$) before being classified. However, this approach assigns the same weight to each step ($1/T$), ignoring any interdependence between them, for example, if the previous time step has already produced a spike, it may be more difficult for the current time step to produce a new spike again, so the signal from the new spike generated by the current time step may be stronger. This idea is not considered in cases where different time steps are given the same weight, which can lead to reduced performance.

To address this, our approach employs learnable parameters to replace the fixed averaging weights. We incorporated this modification into the Spikingformer (Zhou et al., 2023a;b). For benchmarking purposes, we also examined two sets of fixed parameters: one based on arithmetic differences (**AD**)

and another based on arithmetic ratios (**AR**). Experimental outcomes corroborate the efficacy of our proposed Time-Steps Weight (**TSW**) mechanism (As shown in Table 6).

Table 6: Effect of TSW on model performance. The best results on each dataset have been highlighted in bold font.

| Acc | Baseline | AD | AR | TSW |
|---|---|---|---|---|
| **CIFAR 10** | 95.93 | 96.00 | 96.03 | **96.07** |
| **CIFAR 100** | 79.64 | 79.67 | 79.67 | **79.70** |

## A.2 IMPLEMENTATION DETAILS OF THE MAIN EXPERIMENT

We used the openai/clip-vit-base-patch16 [2] from Huggingface as the pre-trained CLIP model, which has a dimension of 512. We used a Spikingformer-4-384((Zhou et al., 2023b)) with 4 layers and a dimension of 384 as the base model for comparison. The image-side component architecture of SpikeCLIP is built upon a spikingformer-4-384 base and incorporates a time-step weight (TSW) layer followed by a dimensionality-mapping layer, aligning the output to a 512-dimensional space compatible with pre-trained CLIP models.

In order to compare with SpikeCLIP, we constructed an ANN counterpart of SpikeCLIP, ScratchCLIP. ScratchCLIP's image encoder is a 4-layer Transformer architecture, and it uses a patch splitting layer with the same number of parameters as SpikeCLIP (the patch splitting layer of conventional CLIP has only one convolution layer with fewer parameters), and ScratchCLIP's text encoder uses an MLP architecture, as well as a word embedding layer of conventional CLIP. Like SpikeCLIP, ScratchCLIP's dimension is 384 dimensions and maps images or text to 512-dimensional representations at the output layer.

For SpikeCLIP, we set the threshold of the common spiking neuron $U_{thr}$ to 1.0, and the threshold of the spiking self-attention block neuron to 0.25. In addition, we set the decay rate $\beta = 0.9$, the scaling factor $\tau$ as 0.125, and the time step $T$ of the peak input of all datasets to 4. In image-side pre-training, we set input dimensions to 224x224 to parallel the pre-trained CLIP model for both CLIP and SpikeCLIP evaluations. To optimize SpikeCLIP's training speed, images were resized to 32x32 using bilinear interpolation. For text-side pre-training, a fixed text length of 20 was employed. We completed the experiment on two devices, each equipped with 4 NVIDIA GeForce RTX 3090 GPUs.

During the pre-training of the SpikeCLIP image encoder, we set the batch size to 196, $\alpha$ to 1, and the learning rate to 5e-3 which remained unchanged after the cosine decay to 5e-4 within 50 epochs. In the pre-training of the SpikeCLIP text encoder, we set the batch size to 1024 and trained 400 epochs. In the fine-tuning, the batch size was 196 and the learning rate was 5e-4 which remained unchanged.

## A.3 ANALYSIS OF THE TEXT ENCODER ARCHITECTURE OF SPIKECLIP

To draw a comparison with the Contrastive Language-Image Pretraining (CLIP) model, we initially employed a Transformer-based architecture for the text encoder, which is analogous to the architecture used for the image encoder. This was trained on our newly constructed dataset $D_{txt}$. However, we observed that this Transformer-based text encoder struggled with effective loss minimization during training and also demonstrated poor accuracy when integrated with the image encoder. An improvement was noted upon switching to a Multilayer Perceptron (MLP)-based architecture for the text encoder. Our findings suggest that within the framework of "Alignment Pre-training + Dual-Loss Fine-tuning", the text encoder is prone to overfitting when trained on newly constructed datasets, particularly if the architecture is overly complex. Although the text encoder based on MLP architecture is simple in architecture, it is sufficient for the training task applied in our work. Comprehensive experimental results are presented in Table 7 and Figure 4.

In the text encoder architecture, we follow the work of Bal & Sengupta (2023) and use the LayerNorm layer as one of the components of text information processing, on the one hand, because the LayerNorm layer is indeed indispensable in text data processing, on the other hand, from the

---

[2] https://huggingface.co/openai/clip-vit-base-patch16

perspective of hardware design, the design of the chip should be based on the design of relevant algorithms. As the wide application of the BatchNorm layer in SNN architecture is inseparable from the algorithm research of SNN in the field of image processing.

Table 7: Classification results of the text encoders of both architectures on downstream datasets (along with the image encoder of conventional CLIP).

| Architecture | CIFAR 10 | CIFAR 100 | Caltech 101 | Flowers 102 | OxfordIIITPet | STL 10 | Avg |
|---|---|---|---|---|---|---|---|
| Transformer-based | 86.37 | 48.03 | 75.78 | 27.09 | 33.93 | 94.76 | 60.99 |
| MLP-based | **90.63** | **64.69** | **79.88** | **62.86** | **81.79** | **97.58** | **79.57** |

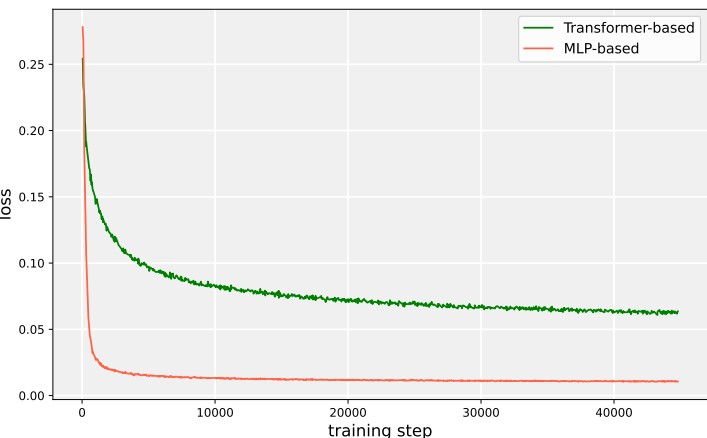

Figure 4: The change of loss value during the training of two text-side architectures.

## A.4 DATASET

The datasets employed across the aforementioned experiments are delineated below:

- **ImageNet-1k**: The ImageNet-1k serves as a foundational benchmark in computer vision research, comprising approximately 1.2 million high-resolution color images across 1,000 distinct categories. The dataset is commonly partitioned into training, validation, and testing subsets to enable rigorous evaluation of machine learning models. Due to its scale and diversity, ImageNet-1k has become instrumental in the development and assessment of state-of-the-art algorithms. In addition, this dataset is one of the largest image classification datasets available(Russakovsky et al., 2015).
- **CIFAR10**: The CIFAR10 serves as a well-established benchmark within the domains of machine learning and computer vision. Comprising 60,000 color images with a resolution of 32x32 pixels, the dataset is organized into 10 unique classes. With each class containing 6,000 images, the dataset ensures a balanced class distribution. Conventionally, CIFAR10 is partitioned into 50,000 images for training and 10,000 images for testing, thereby providing a consistent framework for evaluating the performance of classification models(Krizhevsky, 2009).
- **CIFAR100**: An extension of the CIFAR10 dataset, CIFAR100 is also a prominent benchmark in the fields of machine learning and computer vision. While maintaining the same overall count of 60,000 color images at a 32x32 pixel resolution, CIFAR100 expands the class diversity to 100 distinct categories, each represented by 600 images. For evaluative purposes, the dataset is typically segmented into 50,000 training images and 10,000 testing images. This augmented class variety enhances CIFAR100's utility for conducting more nuanced assessments of classification models(Krizhevsky, 2009).

- **Flower102**: The Flower102 dataset is a notable asset within the computer vision landscape, explicitly designed to cater to fine-grained image recognition endeavors. The dataset comprises a diverse set of images, capturing 102 different floral species. Each category is scrupulously curated to maintain a balanced representation, thereby enabling more sophisticated model evaluations. Due to its focus on capturing subtle variances between closely aligned classes, the Flower102 dataset plays a pivotal role in both refining and benchmarking specialized image classification algorithms(Nilsback & Zisserman, 2008).
- **Caltech101**: As an esteemed benchmark in computer vision research, the Caltech101 dataset encompasses an assemblage of approximately 9,000 color images, categorized into 101 distinct object classes. These classes span a diverse array of subjects, including animals, vehicles, and inanimate objects, with a fluctuating number of images allocated to each category. Widely employed for a variety of computational tasks, such as object recognition and classification, Caltech101 offers a multifaceted visual dataset for the rigorous evaluation of machine learning model performance(Fei-Fei et al., 2004).
- **OxfordIIIPet**: The OxfordIIIPet dataset holds a significant position in the realm of computer vision, particularly in the context of fine-grained classification assignments. The dataset comprises visual representations of 37 distinct breeds of cats and dogs, furnishing a nuanced foundation for algorithms engineered to discern subtle visual cues. Each breed category is populated with a balanced assortment of images, thereby facilitating the compilation of representative training and testing subsets. Owing to its targeted emphasis on the classification of pet breeds, the OxfordIIIPet dataset proves invaluable for fine-tuning models aimed at specialized image recognition tasks(Parkhi et al., 2012).
- **STL10**: The STL10 dataset is characterized by its collection of color images with a 96x96 pixel resolution, and it includes 10 unique categories that parallel those found in the CIFAR10 dataset. It is organized into distinct segments: a labeled set that consists of 5,000 images, an unlabeled set with 100,000 images, and an 8,000-image test set reserved for evaluation. This configuration provides a versatile framework for both supervised and unsupervised learning approaches, making it a useful resource for a diverse array of machine-learning applications.
- **D-text**:
  For the purpose of training the text encoder, we curated a dataset comprising 115,708 textual entries derived from the labels of 27 datasets used in CLIP's zero-shot evaluation, along with their respective templates.
  To elucidate, let's consider the CIFAR10 dataset as an example: with its 10 labels and 18 associated templates, it contributes to the formation of D-text by generating 180 distinct text segments[3].
  Here are a few of the templates of CIFAR10:
    - *A blurry photo of a {}.*
    - *A black and white photo of a {}.*
    - *A high-contrast photo of a {}.*
    - *A photo of a big {}.*

## A.5 CROSS-MODAL IMAGE CLASSIFICATION

To assess the modal alignment capabilities of SpikeCLIP, we designed two distinct experimental paradigms aimed at evaluating its classification ability. The first approach involved *Unseen Label Set*. Using the CIFAR10 dataset as a representative example, for each label within CIFAR10, we replaced it with the closest analogous label from the CIFAR100 and ImageNet-1k datasets. The selection process was facilitated through a specific prompt, termed *Prompt1*, with the assistance of ChatGPT (OpenAI, 2022). Additionally, we conducted four sub-experiments involving random label replacement at different scales: specifically 20%, 40%, 80% and 100%. For the initial three scenarios, predefined random seeds were used, and each was executed in triplicate to record both the *mean* and *variance* of the results.

The second experimental paradigm focused on *Expanded Label Set*. Once again employing the CIFAR10 dataset, we used a separate prompt, *Prompt2*, to engage ChatGPT in the selection of $N \times 10$ labels that were most dissimilar to the original 10 labels of CIFAR10. This effectively

---

[3]https://github.com/openai/CLIP

expanded the label set by a factor of $(N + 1)$. Subsequently, classification accuracy was evaluated under these modified conditions.

- **Prompt1:** The following is the label list *L1* for dataset *DS1*. Please select the label that is closest to label $x$: *L1*.

- **Prompt2:** The following are the label lists for dataset *DS0*, *L0*, and dataset *DS2*, *L2*. Please select $N$ labels from *L1* that are the least similar to the labels in *L0*: *L0*, *L2*.

In the above Prompts, $DS0 \in$ {CIFAR10, STL10}, $DS1 \in$ {CIFAR100, ImageNet-1k}, and $DS2 \in$ {CIFAR100}.

## A.6 THE IMPACT OF DATASET SIZE AND DATA DISTRIBUTION

Owing to limitations in acquiring a large dataset of image-text pairs, our SpikeCLIP model was unable to undergo the same pre-training regimen as the original CLIP model. Nonetheless, we posit that with access to adequate training data, SpikeCLIP's performance can be enhanced. To substantiate this hypothesis, we designed a specific experimental setup.

We use two metrics to quantify the amount of training data: data volume and data distribution. The term data volume refers to the total number of samples utilized during training, while data distribution denotes the level of similarity between the training and evaluation data. Our experiments employ two evaluation datasets: CIFAR10 and ImageNet-1k. For instance, when conducting evaluations on CIFAR10, we set six different levels of training data volume, ranging from 0k to 100k. Regarding data distribution, we establish three different dataset mixing schemes with varying levels of similarity to CIFAR10, detailed as follows, where the size of the data volume is denoted as $N$:

- **For evaluations on CIFAR10:**
  - **Slightly-similar:** $\frac{1}{3}N$ CIFAR10 + $\frac{1}{3}N$ CIFAR100 + $\frac{1}{3}N$ ImageNet-1k;
  - **Intermediate:** $\frac{1}{2}N$ CIFAR100 + $\frac{1}{2}N$ ImageNet-1k;
  - **Dissimilar:** Only ImageNet-1k.
- **For evaluations on ImageNet-1k:**
  - **Slightly-similar:** $\frac{1}{3}N$ ImageNet-1k + $\frac{1}{3}N$ CIFAR100 + $\frac{1}{3}N$ CIFAR10;
  - **Intermediate:** $\frac{1}{2}N$ CIFAR100 + $\frac{1}{2}N$ CIFAR10;
  - **Dissimilar:** Only CIFAR10.

## A.7 ENERGY CONSUMPTION

According to Yao et al. (2022), for SNNs, the theoretical energy consumption of layer $l$ can be calculated as:

$$Energy(l) = E_{AC} \times SOPs(l) \tag{7}$$

where SOPs is the number of spike-based accumulate (AC) operations. For classical ANNs, the theoretical energy consumption required by the layer $b$ can be estimated by:

$$Energy(b) = E_{MAC} \times FLOPs(b) \tag{8}$$

where FLOPs is the floating point operations of $b$, which is the number of multiply-and-accumulate (MAC) operations. We assume that the MAC and AC operations are implemented on the 45nm hardware (Horowitz, 2014), where $E_{MAC} = 4.6pJ$ and $E_{AC} = 0.9pJ$ (1J $= 10^3$ mJ $= 10^{12}$ PJ). The number of synaptic operations at the layer $l$ of an SNN is estimated as:

$$SOPs(l) = T \times \gamma \times FLOPs(l) \tag{9}$$

where $T$ is the number of time steps required in the simulation, $\gamma$ is the firing rate of the input spike train of the layer $l$.

Therefore, we estimate the theoretical energy consumption of SpikeCLIP as follows:

$$E_{SpikeCLIP} = E_{AC} \times \left( \sum_{m=1}^{M} \text{SOP}_{\text{SNN FC}}^{m} + \sum_{n=1}^{N} \text{SOP}_{\text{SNN Conv}}^{n} \right) \tag{10}$$

where SNN FC, and SNN Conv are the fully connected linear layer and convolutional layer with neurons in SpikeCLIP, respectively. As shown in formula 10, the SOPs of $m$ SNN Fully Connected Layer (FC), $n$ SNN Convolutional layers are added together and multiplied by $E_{AC}$.

We refer to Horowitz (2014), assuming that MAC and AC operations are implemented on 45nm hardware (the calculation of power consumption in this hardware only involves MAC and AC operations) since SpikeCLIP and ScratchCLIP have the same architecture except for pulsar neurons, We can calculate the energy consumption reduction (ECR) by formulas 7, 8, 9 and 10 as the following expression formula:

$$ECR = 1 - \frac{E_{AC} \times T \times \bar{\gamma}}{E_{MAC}} \tag{11}$$

where $E_{MAC} = 4.6pJ$, $E_{AC} = 0.9pJ$, $\bar{\gamma}$ represents the average neuron firing rate of the whole SpikeCLIP.

