# OpenReview forum: "SpikeCLIP: A Contrastive Language-Image Pretrained Spiking Neural Network"
_ICLR.cc/2024/Conference — ICLR 2024 Conference Withdrawn Submission_

### Official Review · Reviewer_EDPh · 2023-10-24

**Soundness:** 2 fair
**Presentation:** 2 fair
**Contribution:** 1 poor
**Rating:** 3
**Confidence:** 4

**Summary:**

The author proposed a cross-modal SNN, named SpikeCLIP, which can perform feature extraction and alignment across multiple modalities.

**Strengths:**

1. The author proposed a cross-modal model for SNN and explored some downstream tasks based on it.

**Weaknesses:**

1. The author mainly adopted the idea of knowledge distillation to train SpikeCLIP, however, a similar scheme [1] has been proposed previously. In addition, regarding the algorithm and neuron model design of SNN, I think the contribution of this paper is very limited. I think this paper is more about directly transferring the concepts related to CLIP to the field of SNN and lacks technical contributions related to SNN.

2. The performance of SpikeCLIP on downstream datasets (CIFAR-10, CIFAR-100) is not superior, and the author did not fully list recent works about single-modality SNN in Table 1. For example, [2] can achieve higher performance (CIFAR-10: 95.58%, CIFAR-100: 78.71%, 4 time-steps) by directly training on ResNet-19 than SpikeCLIP, which means that the author's pre-trained multi-modality model based on ImageNet-1k is even inferior to a single-modality SNN through direct training on ResNet-19.

3. The training dataset (ImageNet-1k) and model parameter size (56.87M) used by the author in this paper are too small. Although this paper is a preliminary exploration of SNN cross-modal learning, the performance achieved by the author and the number of downstream tasks attempted have serious deficiencies.

4. In Figure 2, the text encoder section of SpikeCLIP uses the LayerNorm layer. However, the floating-point multiplication operations involved in LayerNorm are usually not allowed in the inference stage of SNN.

[1] Xu, Qi, et al. "Constructing deep spiking neural networks from artificial neural networks with knowledge distillation." Proceedings of the IEEE/CVF Conference on Computer Vision and Pattern Recognition. 2023.

[2] Duan, Chaoteng, et al. "Temporal effective batch normalization in spiking neural networks." Advances in Neural Information Processing Systems 35 (2022): 34377-34390.

**Questions:**

See Weakness Section.

---

> ### Author Response · Authors · 2023-11-12
>
> **Thank you very much for your valuable comments!**
>
> **Q1: The work of training SNNs using KD principles has already been done, and SpikeCLIP lacks algorithms and neuron designs specific to SNNs, so is the contribution of SpikeCLIP relatively limited?**
>
> **R1:** The work you mentioned [1] employed KD principles to train single-modal CNN-based SNNs in a step-wise manner, whereas our work effectively trained multi-modal Transformer-based SNNs using the paradigm of aligned pre-training followed by dual-loss fine-tuning. The difficulties and emphasis between the two works are different. Due to the scarcity of SNNs in the field of natural language processing (mainly due to the absence of authoritative architectures), we attempted to represent both modalities' information effectively through direct training with spike signals, aligning and fusing them, which is more challenging than distilling single-modal information. To achieve this, we not only modified the original Spikingformer on the image encoder side (adding TSW, as shown in A.1) but also compared the effectiveness of different text encoder architectures. While it is indeed feasible to easily transition from CLIP to the SNN domain using methods such as weight transfer, our focus is on validating whether modal information represented by spike signals can effectively undergo modal fusion, similar to what ANNs achieve. Our work's contribution lies in emphasizing this point.
>
>
>
>
> **Q2: The performance of SpikeCLIP on downstream datasets is not superior, and the author did not fully list recent research achievements in single-modal SNNs in Table 1?**
>
> **R2:** We have included several representative works of single-modal SNNs in Table 1, but not all, as there is indeed a considerable amount of relevant research.  We have also listed the latest works on single-modal SNNs, such as [2] (CIFAR10: 95.95%; CIFAR100: 80.37%, 4-time steps), which outperforms the one you mentioned [3] and is currently the best-performing.
>
> Additionally, you noted that the performance of SpikeCLIP is inferior to a single-modal SNN trained directly through ResNet-19. This is inevitable because comparing the accuracy of a single-modal model with a multi-modal model is inherently unfair and meaningless. As SpikeCLIP is the first multi-modal SNN, we intentionally included in Table 1 a fair comparison between the best single-modal and multi-modal models in both the SNN and ANN domains to better assess its performance.  Table 1 compares the accuracy of models on CIFAR10/100.  In the ANN domain, the best single-modal model has an accuracy advantage of 0.68/4.50% over the best multi-modal model, while in the SNN domain, the best single-modal model outperforms SpikeCLIP by 1.47/2.68%. Compared to the gap in ANNs, the gap in SNNs is within an acceptable range, and on CIFAR100, the gap in SNNs is even smaller. We believe this sufficiently reflects the performance level of SpikeCLIP. Furthermore, SpikeCLIP possesses zero-shot learning capabilities, a feature absent in traditional single-modal SNNs.  Faced with new downstream tasks, SpikeCLIP can follow the fine-tuning paradigm without starting training from scratch, a characteristic worth considering. In conclusion, demanding that multi-modal SNNs surpass single-modal SNNs in performance while leveraging advantages unique to single-modal models is the next important challenge.

---

> > ### Author Response · Authors · 2023-11-12
> >
> > **Q3: Are the training dataset and model parameters used in SpikeCLIP too small? Is there a serious shortage of downstream tasks?**
> >
> > **R3:** While inspired by CLIP, it is unfortunate that we couldn't access the massive datasets used to train CLIP. However, ImageNet-1k is one of the largest image-text datasets available to us. Additionally, in Section 4.3, we experimentally verified that the performance of SpikeCLIP could potentially be enhanced with more data involved in pre-training. This compensates for the regrettable inability to use larger-scale datasets for pre-training.
> >
> > Secondly, as indicated in Table 1, our SpikeCLIP, as an SNN, does not have small model parameters; in fact, it has the largest, as it includes both an image encoder and a text encoder. Of course, when compared to other ANNs, SpikeCLIP has fewer parameters. However, even at this relatively smaller parameter count, SpikeCLIP still demonstrates commendable performance, which is encouraging.
> >
> > Finally, we conducted alignment pre-training using ImageNet and employed datasets with six different levels of data granularity for downstream image classification tasks. In comparison to previous single-modal SNNs (such as [4], [5]) that only used MNIST/CIFAR/ImageNet for classification tasks, our work represents the first attempt in the SNN domain to explore a more diverse set of new datasets.
> >
> >
> >
> >
> >
> > **Q4: Is it unreasonable for SpikeCLIP to use LayerNorm in the text encoder?**
> >
> > **R4:** First, since SNN research is currently focused in the field of computer vision, many related algorithms have driven hardware designs consistent with these algorithms, as shown in [6]. Therefore, the BatchNorm layer, which is widely used in computer vision, is considered suitable for SNNs and represents the consensus in the field. However, there is limited work on SNN design for natural language processing (initially explored in A.3 for different text encoder architectures), and the specific hardware design of SNN (including the LayerNorm layer) for text processing has not been prioritized. However, our approach is algorithm-centric, and we adhere to the principle that particular hardware should cater to particular algorithms.
> >
> > Secondly, when considering whether to apply the LayerNorm layer, we draw on the work [7], which also uses the LayerNorm layer as an indispensable component of SNNs when processing text data, and has achieved good results.
> >
> > In conclusion, we consider this problem to be a major challenge in the design of SNN chips in the field of hardware. Our work verifies the validity and feasibility of the efficient fusion of modal information represented by spike signals in SpikeCLIP from an algorithmic point of view.
> >
> > **[1] Xu Q, Li Y, Shen J, et al. Constructing deep spiking neural networks from artificial neural networks with knowledge distillation[C]//Proceedings of the IEEE/CVF Conference on Computer Vision and Pattern Recognition. 2023: 7886-7895.\
> > [2] Zhou C, Yu L, Zhou Z, et al. Spikingformer: Spike-driven Residual Learning for Transformer-based Spiking Neural Network[J]. arXiv preprint arXiv:2304.11954, 2023.\
> > [3] Duan C, Ding J, Chen S, et al. Temporal effective batch normalization in spiking neural networks[J]. Advances in Neural Information Processing Systems, 2022, 35: 34377-34390.\
> > [4] Fang W, Yu Z, Chen Y, et al. Deep residual learning in spiking neural networks[J]. Advances in Neural Information Processing Systems, 2021, 34: 21056-21069.\
> > [5] Zheng H, Wu Y, Deng L, et al. Going deeper with directly-trained larger spiking neural networks[C]//Proceedings of the AAAI conference on artificial intelligence. 2021, 35(12): 11062-11070.\
> > [6] Pei J, Deng L, Song S, et al. Towards artificial general intelligence with hybrid Tianjic chip architecture[J]. Nature, 2019, 572(7767): 106-111.\
> > [7] Bal M, Sengupta A. Spikingbert: Distilling bert to train spiking language models using implicit differentiation[J]. arXiv preprint arXiv:2308.10873, 2023.**

---

> ### Author Response · Authors · 2023-11-17
>
> **Dear commenter EDPh:**
>
> **We greatly appreciate your time and effort in reviewing our work.** We have considered your questions carefully and made the necessary changes.
> Specifically, we have revised section **A.1** to elaborate in more detail on our introduction of TSW, an architectural setup that significantly improves SNNs performance. We have revised **3.3** to express in more detail the special insights and significance of our "alignment pre-training + dual-loss fine-tuning" training framework; We have added **Table 1** to include the baseline you mentioned in the comparison range; Finally, we give our considerations for choosing to use the LN layer in **A.3**. You can also refer to our official review for a more comprehensive response on the motivation and contribution to this study, the training framework, the performance of SpikeCLIP, and the implementation of LN in SNN.
>
> Your expertise and insights are invaluable and we are keen to ensure that our research meets the highest standards. We look forward to hearing more from you and will be happy to answer further questions.

---

> ### Author Response · Authors · 2023-11-21
>
> **Dear Reviewer EDPh:**
>
> We sincerely thank you for taking the time out of your busy schedule to conduct a thorough review of our paper and provide valuable comments.
>
> As the Author-Review Discussion period is drawing to a close with only **two days** remaining, we would like to ensure that all your concerns have been adequately addressed. If there are any questions or unresolved issues, we are eager to provide further clarification or make necessary revisions.
>
> Best regards,
>
> The Authors

---

### Official Review · Reviewer_njnp · 2023-10-29

**Soundness:** 2 fair
**Presentation:** 2 fair
**Contribution:** 2 fair
**Rating:** 5
**Confidence:** 5

**Summary:**

In the paper, the authors introduce SpikeCLIP inspired by CLIP. To realize the SpikeCLIP, the authors provide an Alignment Pre-training + Dual-Loss Finetuning method. Extensive experiments demonstrate that SNNs achieve comparable results to their DNN counterparts while significantly reducing energy consumption across various datasets commonly used for multimodal model evaluation.

**Strengths:**

1. This is the first work to transfer the CLIP to the SNN field.
2. The authors provide the code, which is good.

**Weaknesses:**

1. The novelty is limited. The two steps can be seen as the KD method. So the work just uses a KD method to convert a CLIP as SpikeCLIP.
2. The results are not good. For image classification, the accuracy is worse than other SOTA methods. For the zero-shot task, since SpikeCLIP is not trained on a really large dataset, it is much worse than CLIP, thus the value of SpikeCLIP is limited, considering that the greatest value of CLIP is suitable for zero-shot tasks.

**Questions:**

see weakness.

---

> ### Author Response · Authors · 2023-11-12
>
> **Thank you very much for your valuable comments!**
>
> **Q1: The novelty seems limited; is the alignment pre-training + dual-loss fine-tuning just a knowledge distillation (KD) method?**
>
> **R1:** Currently, most research on SNNs is primarily concentrated in the field of computer vision, with an extremely limited presence in natural language processing and a complete absence in the application of SNN architectures in the multimodal domain. There is currently no existing work exploring the feasibility of SNN architectures in the multimodal domain and whether the modality information represented by pulse signals can fuse modalities as effectively as the floating-point information in ANNs. However, with the growing prevalence of multimodal applications and the enormous energy consumption pressure brought by ANNs, the application of SNN architectures in the multimodal domain is becoming inevitable. Our work, inspired by CLIP, is the first attempt to explore the feasibility and effectiveness of using the spike-based computing paradigm for modality fusion, urging researchers to pay more attention to the application of SNNs in the multimodal context.
> We propose the alignment pre-training + dual-loss fine-tuning framework, which effectively trains the first competitive and zero-shot learning-capable multimodal SNN.
>
> While our framework draws inspiration from CLIP and incorporates the idea of knowledge distillation (KD), it successfully accomplishes the task on its own merit. Knowledge distillation is a universally effective concept, and the focus of our alignment pre-training + dual-loss fine-tuning framework is not on whether KD principles are used, but on how they are applied. In our paper, we argue that the initial alignment pre-training step enables SpikeCLIP to acquire general image/text representation capabilities. In the dual-loss fine-tuning stage, as described in Section 4.3, we determine the use of two different losses by discerning the impact at different data granularities.
> Moreover, SpikeCLIP is not merely a transformation of CLIP into SpikeCLIP. SpikeCLIP and CLIP have distinct architectures (beyond the SNNs vs. ANNs difference) and different training processes, among other differences. Since CLIP conducts representation comparison at the end of a dual-stream architecture when performing multimodal tasks, a condition also achievable by current SNNs, we chose CLIP as a guide for SpikeCLIP. SpikeCLIP is not presented as a direct transformation from CLIP, and perhaps a better "conversion" could be achieved through weight transfer. However, this is not the focus of our paper.
>
>
>
>
> **Q2: SpikeCLIP's performance is not good; Its accuracy is lower than other state-of-the-art (SOTA) methods, and it is much worse than CLIP.  Does SpikeCLIP have limited value?**
>
> **R2:** Without a doubt, SpikeCLIP's accuracy is expected to be lower than the current best single-modal SNNs and significantly worse than CLIP. However, this comparison is unfair and meaningless because, at the same level, single-modal models perform worse than multimodal models, and under the same conditions, SNNs perform worse than ANNs.
>
> At the same level, multimodal models have zero-shot learning capabilities, a feature we validated in Section 4.2.2 for SpikeCLIP, but currently, all single-modal SNNs lack this capability.
> At the same level, SNNs have lower accuracy than ANNs because, unlike ANNs, SNNs use integer operations to process data streams. While this results in an acceptable decrease in accuracy, it also comes with higher energy efficiency.  In Section 4.5, we demonstrate that SpikeCLIP reduces energy consumption by 77.06% to 78.66% compared to its ANN counterpart.
>
> To more fairly evaluate SpikeCLIP's performance, we compare the performance gaps between the best single-modal and multimodal models in SNNs and ANNs on CIFAR10/100 in Table 1. SpikeCLIP is currently the only multimodal SNN. In Table 1, the performance gap between SpikeCLIP and Spikingformer is 1.47%/2.68%, while the gap between CLIP and ViT is 0.68%/4.50%. These comparisons indicate that, as a multimodal SNN, SpikeCLIP's performance is competitive.
> Of course, it is not our intention to simply discuss the performance of SpikeCLIP. Due to the gap in the topic of multimodal SNN, our work is only a preliminary attempt to determine whether modal information represented by SpikeCLIP can be fused, which is also claimed in our paper. We believe that our work will lead to more outstanding research focusing on the performance weaknesses and application challenges of multimodal SNNs.

---

> ### Comment · Reviewer_njnp · 2023-11-12
>
> Thanks for your response. Considering that all the concerns of reviewers are the same, Maybe I have been a bit strict with the authors, so I have changed my score. However, I agree with other reviewers that the more important thing is actually to find the specifics of spikes in those architectures or settings, not just apply the SNN to other fields. I advise that the authors could give more special insights and meanings for the alignment pre-training + dual-loss fine-tuning framework. About the performance, the performance of the SpikeCLIP on zero-shot tasks is really much worse. I don't think that just SNN can save energy thus its performance can be ignored. So the response can not convince me. In addition, BN can be folded into the weights in the inference, but LN can not be. So the authors' responses to the questions of other reviewers are not good enough.

---

> > ### Author Response · Authors · 2023-11-15
> >
> > **We acknowledge the insightful comments provided by the reviewers and appreciate the opportunity to address their concerns.**
> >
> >
> > **1. The motivation and contribution of this study.**
> >
> > **R1:** There are few works that have demonstrated their effectiveness in natural language processing, particularly in the alignment of feature representations from diverse modalities such as images and texts within an event-based sparse firing regime. Acknowledging the existing gap in the literature regarding the demonstration of SNNs' capacity to align feature representations across modalities and employ them for multi-modal tasks is crucial.
> >
> > Our research is specifically designed to address this gap by investigating the feasibility of leveraging SNNs for feature extraction and alignment in a multi-modal context through event-driven spiking trains. It is imperative to underscore that, despite the inherent challenges, SNNs offer a promising avenue for the implementation of energy-efficient deep neural networks, notably in terms of reduced energy consumption during inference. Our study delves into the practical application of SNNs and provides empirical evidence supporting their ability to achieve results comparable to traditional artificial neural networks. Importantly, we demonstrate a significant reduction in energy consumption across diverse datasets commonly utilized for evaluating multimodal models.
> >
> >
> > **2. The special insights and meanings for the proposed training framework.**
> >
> > **R2:** It is well-known that SNNs are hard to train for complex tasks, and it remains a great challenge due to the lack of efficient training algorithms, even in a software training environment. Inspired by the training paradigm of CLIP, we also tried to train SpikeCLIP by means of image-text contrast training in the pre-experiment stage, but we found that SpikeCLIP could not effectively learn the representation of the two modalities under this setting, and even had the problem of **"gradient disappearance or gradient explosion"**. This is usually caused by **"self-accumulating dynamics"** due to backpropagation through time (**BPTT**) algorithms. To solve this problem, our proposed solution introduces a two-step training strategy, namely “alignment pre-training + dual-loss fine-tuning.”
> >
> > In the alignment pre-training phase, we leverage a novel approach by distilling the knowledge of feature extractions and predictive power from CLIP into spiking-based architectures. This not only speeds up the training process compared to random initialization or training from scratch on large datasets but also leads to improved results. We extend the conventional knowledge distillation (**KD**) method by demonstrating its efficacy in achieving multi-modal feature alignment, even when the teacher model operates on continuous values while the student model utilizes discrete spikes for computation and information transmission. In contrast training, SpikeCLIP is asked to align two discrete spike signals, which is extremely difficult (because of the high information loss in the spiking mode); In our alignment pre-training, the continuous and floating-point representation generated by CLIP can be regarded as a **"bridge"**, which more efficiently and quickly realizes the alignment of **"discrete spikes --> continuous representation --> discrete spikes"**, which is also the reason why we choose to use KD method.
> >
> > During the fine-tuning phase, we use a dual-loss mechanism to enhance the stability of the training process and better preserve the generalization power derived from CLIP. Among them, the cross-entropy loss guarantees the consistency of SpikeCLIP with the real labels. On this basis, because of the loss caused by the inherent spike signals of SpikeCLIP, we ensure the consistency of SpikeCLIP with the task-specific fine-tuned CLIP by applying the KL-divergence loss as a penalty. This setup was inspired by [1] and [2], and the results in Table 3 also show the effectiveness of our double-loss fine-tuning approach. Through these two specially designed training steps, our proposed SpikeCLIP demonstrates competitive performance compared to existing single-modal SNNs while effectively overcoming the inherent constraints associated with fixed label sets in image classification.

---

> > > ### Author Response · Authors · 2023-11-15
> > >
> > > **3. The performance of SpikeCLIP.**
> > >
> > > **R3:** SNNs still lag behind ANNs in terms of accuracy. Through intensive research on SNNs in recent years, the performance gap between deep neural networks (DNNs) and SNNs is constantly narrowing. SNNs cannot currently outperform DNNs on the datasets that were created to train and evaluate conventional DNNs (they use continuous values). Such data should be converted into spike trains by spiking neurons before it can be fed into SNNs, and this conversion might cause a loss of information and result in a reduction in performance. Therefore, the comparison is indirect and unfair. In our study, we have conscientiously chosen existing SNNs as baselines for evaluation to provide a fair and relevant benchmark.  As illustrated in Table 1, SpikeCLIP exhibits accuracy rates of **94.48%** and **77.69%** on CIFAR10 and CIFAR100, respectively. These results surpass most of the previous single-modal SNN, and the performance of SpikeCLIP only decreases by **1.47%** and **2.68%** compared with the current optimal single-modal SNN, Spikingformer[3]. It is worth noting that single-modal and task-specific models typically outperform multi-modal models. Similarly, The traditional CLIP, designed to excel in zero-shot situations, was not optimized for achieving state-of-the-art performance on a specific dataset. In addition, the performance gap of **2.68%** between SpikeCLIP and the state-of-the-art single-modal Spikingformer is substantially smaller than the **4.50%** gap between CLIP and its single-modal counterpart (i.e., Vit) on the CIFAR 100 dataset.   This comparison underscores the reasonably good performance of our model.

---

> > > > ### Author Response · Authors · 2023-11-15
> > > >
> > > > **4. The implementation of BN/LN in SNNs.**
> > > >
> > > > **R4:** Admittedly, the current research on the use of the LayerNorm layer in SNNs architecture is still lacking. We appreciate the reviewers for raising this issue with a very professional eye. However, we use the LN layer in this paper for the following reasons:
> > > >
> > > > **a.**	We always believe that algorithm design promotes the design of relevant hardware that fits the algorithm. Just as the BN layer can be widely used in SNN architecture, this is inseparable from the progress of SNN algorithm design in the image processing field.
> > > >
> > > > However, the research on SNN architecture in the field of text processing is very few, which is undoubtedly one of the reasons why the LN layer cannot achieve the same status as the BN layer in SNN architecture at present. However, we believe that the current incompatibility of the LN layer in SNN architecture does not mean that it will be so in the future, and our work also promotes the adaptation of LN in SNN architecture to a certain extent.
> > > >
> > > > **b.**	The importance of the LN layer for text processing is self-evident. We also tried to use the BN layer to replace the LN layer in the pre-experimental stage, but this was not interpretative.
> > > >
> > > > The work of [4] uses implicit differentiation to train a version of BERT in SNNs, which also explicitly uses the LN layer as one of the indispensable components. Their views on the LN layer are consistent with ours.  The irreplaceability of the LN layer in text processing prompted us to first try to build an SNN architecture that can successfully complete the task of text processing from an algorithmic perspective.
> > > >
> > > > **c.**	Although our work ultimately chose LN as one of the components of the SpikeCLIP text encoder from an algorithmic perspective, we also explored the possibility of using the LN layer from a hardware perspective before that. For example, one of our hopes is the hardware-implementable NeuNorm method in [5]. NeuNorm method fuses normalization operations on the spike neurons (also LIF neurons). In our SpikeCLIP architecture shown in Figure 2, the spike neurons follow the LN layer, which can be integrated into a hardware component according to this method. In addition, the NeuNorm method is different from BN in that it normalizes operations on the batch dimension while normalizing operations on the channel dimension. For the input text data of the shape B×L×D (B is the batch size; L is the sentence length, which should be fixed; D is the dimension), reshaping it into B×D×L and using the NeuNorm method may simulate the function of the LN layer.
> > > >
> > > >
> > > > **Finally**, our work is actually to study from the algorithm perspective whether the modal information represented by spike signals can be effectively fused and show the ability that the current single-modal SNNs do not have (such as zero-shot learning ability). The adaptation of the LN layer in SNNs is also a major challenge for SNNs to attack text processing. Thank you for your attention to this issue, and we hope that our work can encourage researchers to think more deeply about this hardware design topic.
> > > >
> > > >
> > > > **[1] Kingma D P, Welling M. Auto-encoding variational bayes[J]. arXiv preprint arXiv:1312.6114, 2013.**\
> > > > **[2] Zhu B, Niu Y, Han Y, et al. Prompt-aligned gradient for prompt tuning[C]//Proceedings of the IEEE/CVF International Conference on Computer Vision. 2023: 15659-15669.**\
> > > > **[3] Zhou C, Zhang H, Zhou Z, et al. Enhancing the Performance of Transformer-based Spiking Neural Networks by Improved Downsampling with Precise Gradient Backpropagation[J]. arXiv preprint arXiv:2305.05954, 2023.**\
> > > > **[4] Bal M, Sengupta A. Spikingbert: Distilling bert to train spiking language models using implicit differentiation[J]. arXiv preprint arXiv:2308.10873, 2023.**\
> > > > **[5] Wu Y, Deng L, Li G, et al. Direct training for spiking neural networks: Faster, larger, better[C]//Proceedings of the AAAI conference on artificial intelligence. 2019, 33(01): 1311-1318.**

---

> ### Author Response · Authors · 2023-11-17
>
> **Dear commenter njnp:**
>
> **We greatly appreciate your time and effort in reviewing our work.** We have considered your question carefully and made the necessary changes. Specifically, we have revised section **3.3** to elaborate in more detail on the specific insights and implications of our training framework. We have also revised **4.2.1** so that everyone can more clearly compare SpikeCLIP's performance from a fair point of view. Finally, we detail our considerations for choosing the LN layer in section **A.3**.
>
> Your expertise and insights are invaluable and we are keen to ensure that our research meets the highest standards. We look forward to hearing more from you and will be happy to answer further questions.

---

> ### Author Response · Authors · 2023-11-21
>
> **Dear Reviewer njnp:**
>
> We sincerely thank you for taking the time out of your busy schedule to conduct a thorough review of our paper and provide valuable comments.
>
> As the Author-Review Discussion period is drawing to a close with only **two days** remaining, we would like to ensure that all your concerns have been adequately addressed. If there are any questions or unresolved issues, we are eager to provide further clarification or make necessary revisions.
>
> Best regards,
>
> The Authors

---

### Official Review · Reviewer_edZQ · 2023-10-30

**Soundness:** 3 good
**Presentation:** 3 good
**Contribution:** 3 good
**Rating:** 6
**Confidence:** 5

**Summary:**

This paper proposes SpikeCLIP, an image-text multi-modal SNN based on CLIP, and a two-stage training method to fine-tune it on downstream tasks. The resulting model can achieve comparable performance on mainstream image datasets with reduced energy consumption and maintains robustness on zero-shot classifications.

**Strengths:**

1. This paper presents the SNN image-text multi-modal model.

2. The two-stage fine-tuning approach retains the performance of the original CLIP model on various tasks, including classification with undefined class labels.

**Weaknesses:**

1.The proposed model architecture for each modal separately is not innovative enough. The image encoder uses an existing SNN architecture (Spikingformer), while the text encoder is a simpler MLP structure, bypassing the difficulties of processing long sequences with SNNs. This design choice improves training efficiency but may limit the model's text-processing capabilities.

2.The two-stage training process of distillation followed by task-specific fine-tuning lacks specific optimization for SNN computational characteristics.

3.Due to the inaccessibility of CLIP's full pretraining dataset, this work uses the smaller ImageNet-1k for distillation pretraining. This restricts the model's generalization capability compared to the original CLIP, including both image and language modalities. More pretraining data would likely be necessary for the model to serve as a general-purpose multimodal foundation model, which may lead to more future challenges, such as convergence and training efficiency.

**Questions:**

1. Does the simple MLP text encoder sacrifice generalization ability in language understanding? For example, the paper does not describe the text templates used in zero-shot classification. If a fixed template like "a photo of a {label}" is used throughout, the text encoder's role may be oversimplified and insufficient to handle other plausible templates such as “a picture of a {label}”. More details should be provided on text settings.

2. ImageNet-1k is used for alignment pretraining before fine-tuning on other datasets. However, test accuracy after further fine-tuning on ImageNet itself is not reported. Does SpikeCLIP have adequate representational capacity and scalability for such large-scale image classification tasks?

---

> ### Author Response · Authors · 2023-11-12
>
> **Thank you very much for your valuable comments!**
>
> **Q1: Does the MLP text encoder sacrifice the generalization ability of language understanding? Regarding the use of text templates in zero-shot classification.**
>
> **R1:** In order to imbue the text decoder of SpikeCLIP with text representation capabilities and for the purpose of comparison with CLIP, our constructed dataset $D_{txt}$  is derived from 27 datasets used to evaluate CLIP's zero-shot learning ability. As mentioned in A.4, for each dataset, we combine its labels with the corresponding template set, thus constructing  $D_{txt}$ with a total of 115,708 entries. We sincerely apologize for not providing more detailed construction details of  $D_{txt}$ in the paper, causing confusion. However, these template sets encompass various templates, and you can refer to the supplementary material for a detailed examination.
> For the image classification datasets used in this paper, training a text decoder using $D_{txt}$ can yield a sufficiently effective result. However, as you rightly pointed out, a simple MLP text encoder inevitably sacrifices the generalization ability of language understanding, despite our comparison of the effectiveness of two different architectures used as text encoders. Therefore, constructing a better text encoder structure remains an important problem we aim to address in our future work.
>
>
> **Q2: Why is the test accuracy after further fine-tuning on ImageNet itself not reported? Does SpikeCLIP have sufficient representational capacity and scalability?**
>
> **R2:** Following the paradigm of alignment pre-training + dual-loss fine-tuning, we deliberately set the downstream datasets to be different from the pre-training dataset. This setup aims to demonstrate the effectiveness of pre-training and the zero-shot learning capability of SpikeCLIP. In Section A.6, we report SpikeCLIP's performance when the distributions of the pre-training dataset and the downstream dataset are dissimilar, indirectly validating the good performance when using ImageNet-1k for both alignment pre-training and dual-loss fine-tuning.
>
> Furthermore, this is the first attempt to use the spike-based computing paradigm for multimodal classification tasks. Unlike CLIP, we do not have access to larger datasets for alignment pre-training, which indeed imposes significant limitations on SpikeCLIP's representational capacity and scalability, as you rightly mentioned. However, in Section 4.3, we experimentally verify the potential for improved SpikeCLIP performance with the existence of more data. Certainly, this is a challenging task, but it doesn't hinder us from confirming the feasibility of SNNs in multimodal scenarios.

---

> ### Author Response · Authors · 2023-11-17
>
> **Dear reviewer edZQ:**
>
> **We greatly appreciate your time and effort in reviewing our work.** We have considered your questions carefully and made the necessary changes.
> Specifically, we further analyze our considerations for choosing the MLP architecture as the text encoder in section **A.3**.  Although the architecture is simple, in our work, there is also not much text information consisting of labels of data sets, which makes the simple MLP architecture suitable for the text processing capabilities of SpikeCLIP.  In addition, we provide more details of $D_{txt}$ in **A.4** (including different templates), and finally, you can refer to our official review for A more comprehensive summary of SpikeCLIP's performance.
>
> Your expertise and insights are invaluable and we are keen to ensure that our research meets the highest standards. We look forward to hearing more from you and will be happy to answer further questions.

---

> ### Author Response · Authors · 2023-11-21
>
> **Dear Reviewer edZQ:**
>
> We sincerely thank you for taking the time out of your busy schedule to conduct a thorough review of our paper and provide valuable comments.
>
> As the Author-Review Discussion period is drawing to a close with only **two days** remaining, we would like to ensure that all your concerns have been adequately addressed. If there are any questions or unresolved issues, we are eager to provide further clarification or make necessary revisions.
>
> Best regards,
>
> The Authors

---

### Official Review · Reviewer_edTp · 2023-10-31

**Soundness:** 2 fair
**Presentation:** 3 good
**Contribution:** 2 fair
**Rating:** 5
**Confidence:** 5

**Summary:**

This paper proposed SpikeCLIP, a framework inspired by CLIP method to dually deal with both image and text input through spiking neural network. The SpikeCLIP is trained through a two-staged “alignment (CLIP) pre-training + Dual-Loss Fine-tuning”, showing good results in classification tasks and zero-shot learning tasks. Compared to conventional CLIP, spikeCLIP shows theoretically high energy efficiency.

**Strengths:**

- SpikeCLIP is the first multimodal SNN architecture that shows the ability to deal with both text and image input.
- The paper proposes an effective training framework to train SpikeCLIP. The alignment pre-training part finds a good way to transfer the representation learned in CLIP to SpikeCLIP.
- The implementation of the framework and the experiments are solid. Moreover, the experiments results shows robust performance in image classification tasks (including zero-shot setting).

**Weaknesses:**

- Although this is the first paper (as far as I know) to realize spiking version of CLIP, the paper itself is lack of enough novelty. Nowadays, as the surrogate gradient based SNN training methods have been greatly developed, transferring or reproducing a specific architecture in conventional ANN(artificial neural network) to SNN is never a significant issue. More important thing is actually to find the specifics of spikes in those architectures or settings, rather than claiming “we are the first spiking version of xx”. Unfortunately, I did not find such highlights in this paper.
- One may argue that this paper shows the energy efficiency of SpikeCLIP compared with its counterpart ScratchCLIP, however, such comparison is not fair enough. As shown in the appendix, the authors only calculate the SOPs corresponding to each spike, while the updating process of membrane potential, the BN/LN operations are not included. It is worth to note that these are only the computing energy. Moreover, the additional energy that needed to maintain the membrane potential for each neuron, which is actually more severe in reality, is not considered in their calculation. No need to mentioning the factual energy lies mostly in data/weight transferring, which is not mentioned either in the paper (the weight amount is the same, and the data input is in fact more than ANN, given one needs to repeat T times of input). I understand this is a paper focusing on algorithms, discussing the real implementation is somewhat out of the scope. However, if the only novelty or advantage actually is built on energy consuming, such discussion is then unable to be ignored.
- Overall, the performance gap compared to ANN is still very big, and noticing this in only in CIFAR10/100, for larger dataset or more complex tasks, I do believe the gap will be larger.

**Questions:**

Please see above weakness.

---

> ### Author Response · Authors · 2023-11-12
>
> **Thank you very much for your valuable comments!**
>
> **Q1：The paper lacks novelty, SpikeCLIP architecture implementation claims "the first......" However, the bright spot could not be found ?**
>
> **R1:** Our work attempts to align the two modalities with the spiking computing paradigm, which is not done before. As mentioned in the paper, the current implementation methods of SNNs mainly include weight transfer and direct training using surrogate gradients. However, due to the sparsity of data and integer operation in SNN, direct training of SNNs is not easy. Our work has used alignment pre-training and dual-loss fine-tuning to effectively realize the alignment of the two modalities and achieve competitive performance. In previous research on SNNs, the vast majority of work in the field of computer vision has been done, so many questions about architecture design have been conducted around computer vision tasks. However, the lack of work on SNNs in the field of natural language processing makes it difficult to use SNNs for multimodal tasks. Considering that the advanced Spikingformer is based on Transformer, which is derived from natural language processing tasks, we have experimented with different text-side architectures in Transformer and MLP for multimodal classification tasks.
>
> We sincerely apologize for emphasizing "the first..." in the paper. However, due to the lack of exploration in the field of SNNs in the multimodal context, our work represents only a preliminary exploration of the feasibility of this topic. We also hope that in the future, there will be more SNNs-related research that pays close attention to the implementation of multimodal tasks, similar to the attention given to computer vision tasks.
>
>
> **Q2: The comparison of energy efficiency between SpikeCLIP and its counterpart, ScratchCLIP, is unfair, and there are omissions in energy calculations?**
>
> **R2:** To ensure a fair comparison of the performance and energy consumption between SpikeCLIP and its counterpart ScratchCLIP, we configured both with identical settings (e.g., layer number, dimensions, training process) except for the spiking neurons. In this setting, energy consumption calculations were based on [1], where SNN, as a neural morphic computing algorithm, can smoothly execute on sparse neural morphic chips, requiring only spike-based accumulate (AC) operations, while ANNs involve numerous multiply-and-accumulate (MAC) operations. Therefore, the additional energy required for membrane potential update processes and maintaining the membrane potential of each neuron does not need to be considered. Specific chip implementation details can be found in [2] and [3]. Regarding BN/LN operations, as SpikeCLIP and ScratchCLIP architectures have identical BN/LN operations, their energy consumption cancels out when calculating the energy consumption ratio.
>
>  While it is acknowledged that the data input for SNN is greater than that for the corresponding ANN, SNN can leverage this additional input to abandon more precise floating-point calculations, sacrificing precision for efficient energy consumption. SpikeCLIP, in its preliminary attempt at multimodal alignment tasks in SNNs, exhibits a performance gap with the single-modal Spikingformer within a reasonable range. At the same time, Spikeclip also shows zero-shot learning ability (which is not available in previous single-modal SNNs). Superior energy efficiency is only an inherent but indispensable advantage of SpikeCLIP.

---

> > ### Author Response · Authors · 2023-11-12
> >
> > **Q3: The performance gap in SpikeCLIP compared to ANN is very large, and the gap is even greater for larger datasets or more complex tasks ?**
> >
> > **R3:** Of course, to make SNNs better than ANNs in terms of performance, this is a long way to go, and almost all the best SNNs can’t exceed the performance of their counterparts in ANNs. However, SNNs emerged because of their efficient energy utilization, biological plausibility, event-driven nature, and rapid inference capabilities, and it is unfair to simply compare the performance of SNNS with ANNs.
> >
> > At the same time, on CIFAR10/100, as a multimodal SNN, the gap between SpikeCLIP and the optimal single-modal Spikingformer is only 1.47/2.68%, while in ANNs, the gap between the optimal single-modal CLIP and multimodal ViT is as high as 0.68/4.50%. The comparison of these two gaps can more fairly reflect the performance of SpikeCLIP. Secondly, in section 4.3 of the paper, we conducted ablation experiments to verify the possibility that SpikeCLIP's performance could be improved with more data. As for your question that SpikeCLIP's performance would deteriorate under more complex tasks? Since SpikeCLIP is inspired by CLIP, we also followed CLIP to test its zero-shot learning ability when doing image classification tasks. Even compared to previous single-modal SNNs work in computer vision (e.g. [4],[5]), we use far more datasets than they do (these works only used ImageNet/CIFAR/MNIST). Of course, getting better at more complex datasets and tasks is also the next significant goal under this theme.
> >
> > **[1] Yao M, Zhao G, Zhang H, et al. Attention spiking neural networks[J]. IEEE transactions on pattern analysis and machine intelligence, 2023.\
> > [2] Pei J, Deng L, Song S, et al. Towards artificial general intelligence with hybrid Tianjic chip architecture[J]. Nature, 2019, 572(7767): 106-111.\
> > [3] Merolla P A, Arthur J V, Alvarez-Icaza R, et al. A million spiking-neuron integrated circuit with a scalable communication network and interface[J]. Science, 2014, 345(6197): 668-673.\
> > [4] Fang W, Yu Z, Chen Y, et al. Deep residual learning in spiking neural networks[J]. Advances in Neural Information Processing Systems, 2021, 34: 21056-21069.\
> > [5] Zheng H, Wu Y, Deng L, et al. Going deeper with directly-trained larger spiking neural networks[C]//Proceedings of the AAAI conference on artificial intelligence. 2021, 35(12): 11062-11070.**

---

> ### Author Response · Authors · 2023-11-17
>
> **Dear reviewer edTp:**
>
> **We greatly appreciate your time and effort in reviewing our work.** We have carefully considered your questions and made the necessary changes. Specifically, we went deeper into the special insights and implications of our "alignment pre-training + dual-loss fine-tuning" training framework in **3.3**, and we modified section **A.6** to make it clearer how energy consumption compares with ScratchCLIP. Finally, you can refer to our official review for a more comprehensive summary of our motivations and contributions to this study and SpikeCLIP's performance.
>
> Your expertise and insights are invaluable and we are keen to ensure that our research meets the highest standards. We look forward to hearing more from you and will be happy to answer further questions.

---

> ### Author Response · Authors · 2023-11-21
>
> **Dear Reviewer edTp:**
>
> We sincerely thank you for taking the time out of your busy schedule to conduct a thorough review of our paper and provide valuable comments.
>
> As the Author-Review Discussion period is drawing to a close with only **two days** remaining, we would like to ensure that all your concerns have been adequately addressed. If there are any questions or unresolved issues, we are eager to provide further clarification or make necessary revisions.
>
> Best regards,
>
> The Authors

---

### Author Response · Authors · 2023-11-17

**We would like to thank the reviewers for their insights and efforts to provide high-quality reviewers. The following are the common concerns of the reviewers.**

**1. The motivation and contribution of this study.**

**R1:** There are few works that have demonstrated their effectiveness in natural language processing, particularly in the alignment of feature representations from diverse modalities such as images and texts within an event-based sparse firing regime. Acknowledging the existing gap in the literature regarding the demonstration of SNNs' capacity to align feature representations across modalities and employ them for multi-modal tasks is crucial.

Our research is specifically designed to address this gap by investigating the feasibility of leveraging SNNs for feature extraction and alignment in a multi-modal context through event-driven spiking trains. It is imperative to underscore that, despite the inherent challenges, SNNs offer a promising avenue for the implementation of energy-efficient deep neural networks, notably in terms of reduced energy consumption during inference. Our study delves into the practical application of SNNs and provides empirical evidence supporting their ability to achieve results comparable to traditional artificial neural networks. Importantly, we demonstrate a significant reduction in energy consumption across diverse datasets commonly utilized for evaluating multimodal models.

**2. The special insights and meanings for the proposed training framework.**

**R2:** It is well-known that SNNs are hard to train for complex tasks, and it remains a great challenge due to the lack of efficient training algorithms, even in a software training environment. Inspired by the training paradigm of CLIP, we also tried to train SpikeCLIP by means of image-text contrast training in the pre-experiment stage, but we found that SpikeCLIP could not effectively learn the representation of the two modalities under this setting, and even had the problem of **"gradient disappearance or gradient explosion"**. This is usually caused by **"self-accumulating dynamics"** due to backpropagation through time (**BPTT**) algorithms. To solve this problem, our proposed solution introduces a two-step training strategy, namely “alignment pre-training + dual-loss fine-tuning.”

In the alignment pre-training phase, we leverage a novel approach by distilling the knowledge of feature extractions and predictive power from CLIP into spiking-based architectures. This not only speeds up the training process compared to random initialization or training from scratch on large datasets but also leads to improved results. We extend the conventional knowledge distillation (KD) method by demonstrating its efficacy in achieving multi-modal feature alignment, even when the teacher model operates on continuous values while the student model utilizes discrete spikes for computation and information transmission. In contrast training, SpikeCLIP is asked to align two discrete spike signals, which is extremely difficult (because of the high information loss in the spiking mode); In our alignment pre-training, the continuous and floating-point representation generated by CLIP can be regarded as a **"bridge"**, which more efficiently and quickly realizes the alignment of **"discrete spikes --> continuous representation --> discrete spikes"**, which is also the reason why we choose to use KD method.

During the fine-tuning phase, we use a dual-loss mechanism to enhance the stability of the training process and better preserve the generalization power derived from CLIP. Among them, the cross-entropy loss guarantees the consistency of SpikeCLIP with the real labels. On this basis, because of the loss caused by the inherent spike signals of SpikeCLIP, we ensure the consistency of SpikeCLIP with the task-specific fine-tuned CLIP by applying the KL-divergence loss as a penalty. This setup was inspired by [1] and [2], and the results in Table 3 also show the effectiveness of our double-loss fine-tuning approach. Through these two specially designed training steps, our proposed SpikeCLIP demonstrates competitive performance compared to existing single-modal SNNs while effectively overcoming the inherent constraints associated with fixed label sets in image classification.

---

> ### Author Response · Authors · 2023-11-17
>
> **3. The performance of SpikeCLIP.**
>
> **R3:** SNNs still lag behind ANNs in terms of accuracy. Through intensive research on SNNs in recent years, the performance gap between deep neural networks (DNNs) and SNNs is constantly narrowing. SNNs cannot currently outperform DNNs on the datasets that were created to train and evaluate conventional DNNs (they use continuous values). Such data should be converted into spike trains by spiking neurons before it can be fed into SNNs, and this conversion might cause a loss of information and result in a reduction in performance. Therefore, the comparison is indirect and unfair. In our study, we have conscientiously chosen existing SNNs as baselines for evaluation to provide a fair and relevant benchmark. As illustrated in Table 1, SpikeCLIP exhibits accuracy rates of **94.48% and 77.69%** on CIFAR10 and CIFAR100, respectively. These results surpass most of the previous single-modal SNN, and the performance of SpikeCLIP only decreases by **1.47%** and **2.68%** compared with the current optimal single-modal SNN, Spikingformer[3]. It is worth noting that single-modal and task-specific models typically outperform multi-modal models. Similarly, The traditional CLIP, designed to excel in zero-shot situations, was not optimized for achieving state-of-the-art performance on a specific dataset. In addition, the performance gap of **2.68%** between SpikeCLIP and the state-of-the-art single-modal Spikingformer is substantially smaller than the 4.50% gap between CLIP and its single-modal counterpart (i.e., Vit) on the CIFAR 100 dataset. This comparison underscores the reasonably good performance of our model.

---

> > ### Author Response · Authors · 2023-11-17
> >
> > **4. The implementation of BN/LN in SNNs.**
> >
> > **R4:** Admittedly, the current research on the use of the LayerNorm layer in SNNs architecture is still lacking. We appreciate the reviewers for raising this issue with a very professional eye. However, we use the LN layer in this paper for the following reasons:
> >
> > **a.** We always believe that algorithm design promotes the design of relevant hardware that fits the algorithm. Just as the BN layer can be widely used in SNN architecture, this is inseparable from the progress of SNN algorithm design in the image processing field.
> >
> > However, the research on SNN architecture in the field of text processing is very few, which is undoubtedly one of the reasons why the LN layer cannot achieve the same status as the BN layer in SNN architecture at present. However, we believe that the current incompatibility of the LN layer in SNN architecture does not mean that it will be so in the future, and our work also promotes the adaptation of LN in SNN architecture to a certain extent.
> >
> > **b.** The importance of the LN layer for text processing is self-evident. We also tried to use the BN layer to replace the LN layer in the pre-experimental stage, but this was not interpretative.
> >
> > The work of [4] uses implicit differentiation to train a version of BERT in SNNs, which also explicitly uses the LN layer as one of the indispensable components. Their views on the LN layer are consistent with ours. The irreplaceability of the LN layer in text processing prompted us to first try to build an SNN architecture that can successfully complete the task of text processing from an algorithmic perspective.
> >
> > **c.** Although our work ultimately chose LN as one of the components of the SpikeCLIP text encoder from an algorithmic perspective, we also explored the possibility of using the LN layer from a hardware perspective before that. For example, one of our hopes is the hardware-implementable NeuNorm method in [5]. NeuNorm method fuses normalization operations on the spike neurons (also LIF neurons). In our SpikeCLIP architecture shown in Figure 2, the spike neurons follow the LN layer, which can be integrated into a hardware component according to this method. In addition, the NeuNorm method is different from BN in that it normalizes operations on the batch dimension while normalizing operations on the channel dimension. For the input text data of the shape B×L×D (B is the batch size; L is the sentence length, which should be fixed; D is the dimension), reshaping it into B×D×L and using the NeuNorm method may simulate the function of the LN layer.
> >
> > **Finally**, our work is actually to study from the algorithm perspective whether the modal information represented by spike signals can be effectively fused and show the ability that the current single-modal SNNs do not have (such as zero-shot learning ability). The adaptation of the LN layer in SNNs is also a major challenge for SNNs to attack text processing. Thank you for your attention to this issue, and we hope that our work can encourage researchers to think more deeply about this hardware design topic.
> >
> > **[1] Kingma D P, Welling M. Auto-encoding variational bayes[J]. arXiv preprint arXiv:1312.6114, 2013.**\
> > **[2] Zhu B, Niu Y, Han Y, et al. Prompt-aligned gradient for prompt tuning[C]//Proceedings of the IEEE/CVF International Conference on Computer Vision. 2023: 15659-15669.**\
> > **[3] Zhou C, Zhang H, Zhou Z, et al. Enhancing the Performance of Transformer-based Spiking Neural Networks by Improved Downsampling with Precise Gradient Backpropagation[J]. arXiv preprint arXiv:2305.05954, 2023.**\
> > **[4] Bal M, Sengupta A. Spikingbert: Distilling bert to train spiking language models using implicit differentiation[J]. arXiv preprint arXiv:2308.10873, 2023.**\
> > **[5] Wu Y, Deng L, Li G, et al. Direct training for spiking neural networks: Faster, larger, better[C]//Proceedings of the AAAI conference on artificial intelligence. 2019, 33(01): 1311-1318.**

---

### Author Response · Authors · 2023-11-23
**To All Reviewers**

We would like to thank the reviewers for their insightful comments and efforts put into providing high-quality reviews.   We plan to release the source codes of SpikeCLIP to the research community.   We appreciate all of you for your comments highlighting the strengths of our work for a summary.

-Our proposed SpikeCLIP first verifies the feasibility and effectiveness of modal alignment and fusion using discrete spike signals.

-Our proposed "alignment pre-training + dual-loss fine-tuning" paradigm effectively solves the problem that it is difficult to train the model in multi-mode scenarios using discrete spike signals.

-As the first multi-modal SNN, SpikeCLIP achieves good performance while showing zero-shot learning ability.


We also sincerely thank reviewers for your constructive feedback and questions to improve our manuscript.  We have addressed the questions raised by reviewers.   If our response addresses your concerns and you would like to accept our paper, would you please raise the scores?   Thank you very much in advance!   **:-D**